JCB Journal of Cell Biology

# STIM1/2 maintain signaling competence at ER-PM contact sites during neutrophil spreading

Camille Rabesahala de Meritens[1], Amado Carreras-Sureda[1], Nicolas Rosa[1], Robert Pick[2], Christoph Scheiermann[2], and Nicolas Demaurex[1]

**Neutrophils are highly motile leukocytes that migrate inside tissues to destroy invading pathogens. $Ca^{2+}$ signals coordinate leukocytes migration, but whether $Ca^{2+}$ fluxes mediated by Stim proteins at ER-PM contact sites regulate neutrophil actin-based motility is unclear. Here, we show that myeloid-specific *Stim1/2* ablation decreases basal cytosolic $Ca^{2+}$ levels and prevents adhesion-induced $Ca^{2+}$ elevations in mouse neutrophils, reducing actin fiber formation and impairing spreading. Unexpectedly, more ER-PM contact sites were detected on the actin-poor adhesive membranes of *Stim1/2*-deficient neutrophils, which had reduced inositol-1,4,5-trisphosphate receptor ($IP_3R$) immunoreactivity on confocal and immunogold micrographs despite preserved $IP_3R$ levels on western blots. Remarkably, *Stim1/2*-deficient neutrophils regained signaling and spreading competence in $Ca^{2+}$-rich solutions and were recruited more effectively in mouse inflamed cremaster muscles in vivo. Our findings indicate that Stim1/2 preserve $IP_3R$ functionality in neutrophils, generating adhesion-dependent $Ca^{2+}$ signals that control actin dynamics during neutrophil spreading. Stim proteins thus maintain $IP_3R$ signaling competence at adhesive membranes, enabling $Ca^{2+}$-dependent actin remodeling during spreading in mouse neutrophils.**

## Introduction

Neutrophils are circulating white blood cells that defend our body against bacterial and fungal infections (Liew and Kubes, 2019). In response to chemotactic cues, neutrophils migrate into tissues via the synergistic activation of chemokine receptors and the engagement of selectins and β2 integrins that mediate neutrophils capture and rolling along the endothelium, followed by their adhesion, polarization, and trans-endothelial migration (Ley et al., 2007; Zarbock and Ley, 2009). The combined activation of G-protein–coupled receptors by chemokines, which signal through PLCβ, and of the tyrosine kinase Syk coupled to phospholipase C (PLC) gamma by selectins generates inositol 1,4,5-trisphosphate ($IP_3$), which opens its cognate receptors ($IP_3R$) on the ER to promote the release of $Ca^{2+}$ from intracellular stores (Hogg et al., 2011; Huang et al., 2017; Mueller et al., 2010). The resulting cytosolic $Ca^{2+}$ elevations enhance the affinity of integrins for adhesion molecules on endothelial cells (Ginsberg, 2014; Kubes, 2002; Ley et al., 2007), which further activate the Syk–PLC gamma signaling pathway to boost $Ca^{2+}$ elevations (Immler et al., 2018; Jakus et al., 2009; Mócsai et al., 2006), strengthening neutrophil adhesion by promoting the formation of integrin clusters on the endothelium (Dixit and Simon, 2012; Ginsberg, 2014; Obergfell et al., 2002; Pardi et al., 1989; Wehrle-Haller, 2006).

Pioneering studies reported both local and global $Ca^{2+}$ elevations associated with the dynamic remodeling of the actin cytoskeleton as neutrophils spread and migrate on substrates in vitro (Jaconi et al., 1991; Kruskal et al., 1986; Petersen et al., 1993). Actin polymerization is initiated at sites of integrin engagement (Sarantos et al., 2008), the polymerized F-actin at pseudopods pulling toward the chemotactic gradient (Filippi, 2019). Local increases in $Ca^{2+}$ at adhesion sites influence the micro-clustering of the integrin LFA-1 (CD11a/CD18; $\alpha_L\beta_2$), strengthening adhesion and F-actin recruitment (Dixit et al., 2011; Lum et al., 2002). In contrast, $Ca^{2+}$ elevations restricted to the uropod facilitate actin depolymerization by releasing integrin/actin complexes (Bengtsson et al., 1993; Downey et al., 1990), enabling selectins and integrins to sling over the cell to form new nascent adhesion points as cells roll (Lum et al., 2002; Sundd et al., 2012). The coordination of local and global $Ca^{2+}$ elevations sustain directed migration and enable neutrophils to search for optimal points to cross the endothelium (Dixit and Simon, 2012). Localized $Ca^{2+}$ elevations also control the formation of phagocytic vacuoles and their fusion with lysosomes by orchestrating the dynamic remodeling of the actin cytoskeleton at phagocytic cups and around vacuoles (Bengtsson et al., 1993; Nunes et al., 2012).

[1]Department of Cell Physiology and Metabolism, University of Geneva, Geneva, Switzerland; [2]Department of Pathology and Immunology, University of Geneva, Geneva, Switzerland.

Correspondence to Nicolas Demaurex: Nicolas.Demaurex@unige.ch.



The maintenance of Ca²⁺ signals generated by IP₃R-mediated Ca²⁺ release require the influx of Ca²⁺ ions into cells to complement the limited amounts of ions stored in intracellular sources. Neutrophils have limited amounts of ER, the major Ca²⁺ store of eukaryotic cells, and might therefore be particularly reliant on Ca²⁺ influx. The main Ca²⁺ entry pathway of immune cells is store-operated Ca²⁺ entry (SOCE) mediated by interactions between stromal interaction molecule (STIM) proteins on the ER and plasma membrane (PM) Ca²⁺-permeable channels of the Orai family (Feske et al., 2005; Liou et al., 2005; Roos et al., 2005; Vig et al., 2006; Zhang et al., 2005). STIM1 and its related isoform STIM2 sense the Ca²⁺ depletion of the ER and translocate to the PM, where they trap and gate Orai channels, triggering the entry of Ca²⁺ into cells to fuel Ca²⁺-dependent effector functions (Park et al., 2009). STIM1-ORAI1 coupling is required for the proliferation of T cells, and patients bearing loss-of-function mutations in *STIM1* or *ORAI1* suffer from severe combined immunodeficiencies (Picard et al., 2009; Vaeth et al., 2020).

The reliance of neutrophil effector functions on STIM-ORAI signaling has been explored in human cell lines and mouse neutrophils using genetic approaches. In neutrophil-like HL-60 cells, STIM1 and ORAI1, but not STIM2, sustain the production of reactive oxygen species (ROS) by the neutrophil NADPH oxidase (Bréchard et al., 2009; Steinckwich et al., 2015). In mouse neutrophils, Stim1 sustains NADPH oxidase activity (Zhang et al., 2014) and enhances phagocytosis by promoting Ca²⁺ microdomains around phagosomes (Nunes et al., 2012), while Stim2 is dispensable for ROS production and phagocytosis but required for cytokine production (Clemens et al., 2017). At odds with these findings, preserved Ca²⁺ fluxes and effector functions were observed in neutrophils from two patients with loss-of-function mutations in *STIM1* and *ORAI1* genes (Elling et al., 2016), possibly reflecting differences in human and mouse neutrophils (Nauseef, 2023). Evidence for a role of STIM proteins in driving neutrophil migration is mixed. *STIM1* silencing reduced *N*-formylmethionyl-leucyl-phenylalanine (fMLP)-induced polarization in HL-60 cells (Bréchard et al., 2009) and myeloid-specific *Stim1* ablation reduced neutrophil migration in vitro and in a psoriasis model of skin infection in mice (Steinckwich et al., 2015). Subsequent studies reported that neutrophils from *Stim1⁻/⁻* chimeric mice adhered and migrated normally in vitro and were efficiently recruited to sites of sterile inflammation in vivo (Sogkas et al., 2015; Zhang et al., 2014). Normal migration was also reported in *Stim2* and *Stim1/2*-deficient mouse neutrophils (Clemens et al., 2017). In contrast, chemotaxis was impaired in neutrophils from *Orai1⁻/⁻* chimeric mice exposed to C5a in vitro and in LPS-induced peritonitis (Sogkas et al., 2015), and *Orai1* haplo-deficiency impaired neutrophil migration under flow by preventing the transition of LFA-1 integrins to the high-affinity conformation required for neutrophil arrest and polarization (Dixit et al., 2011; Schaff et al., 2010). These studies suggest that neutrophil migration rely more on Orai1 than on Stim1 or Stim2, despite the fact that, in mouse neutrophils, *Stim1* ablation consistently decreases SOCE and aborts Ca²⁺ elevations evoked by multiple physiological ligands except C5a (Nunes et al., 2012; Sogkas et al., 2015; Zhang et al., 2014), while *Orai1*

ablation has a minor effect on SOCE (Dixit et al., 2011; Schaff et al., 2010; Sogkas et al., 2015).

To clarify the role of Stim proteins in neutrophil signaling and migration, we generated mice lacking both Stim isoforms and expressing the high-affinity Ca²⁺ indicator Salsa6f in myeloid cells and quantified the Ca²⁺ signals, actin dynamics, and morphological changes as neutrophils spread on solid substrates.

## Results

To record Ca²⁺ signals in primary mouse neutrophils, we crossed Salsa6f transgenic reporter mice (Dong et al., 2017) with mice expressing Cre recombinase under the control of the myeloid-specific promoter CEBPa. Salsa6f comprises a GCaMP6f moiety that responds to Ca²⁺ fluctuations and a tandem dimer red fluorescent moiety (tdT) that serves as reference for ratio Ca²⁺ imaging (Fig. S1 A, sketch). A robust tdT fluorescence signal was detected in cells flushed from the bone marrow (BM) of CEBPa-driven Salsa6f mice, enabling identification of myeloid cells by flow cytometry and microscopy (Fig. 1 A and Fig. S1 A). Approximately 50% of fluorescent cells expressed the granulocyte marker Ly6G (Fig. 1 A) and 87% of tdT/Ly6G⁺ cells were mature neutrophils lacking monocytes and eosinophils markers, a proportion comparable with neutrophils isolated by negative selection (Fig. 1 B). Similar amounts of neutrophils were recovered from mice expressing or not the CRE recombinase (Fig. S1 B), indicating that myeloid expression of Salsa6f does not affect neutrophil development. Post hoc Ly6G staining of Salsa6f-positive cells thus rapidly and efficiently identifies primary neutrophils from BM, increasing 10-fold the amounts of neutrophils that can be recovered by the isolation kit (from 1 to 10 million neutrophils per mouse, with equivalent purity). Salsa6f neutrophils express an endogenous Ca²⁺ reporter and can be immediately used for Ca²⁺ recording modalities, avoiding purification and dye loading procedures that can prime these sensitive cells.

In the absence of chemoattractants, we detected repetitive Ca²⁺ elevations as neutrophils spread and crawled on weakly adhesive poly-L-lysine (PLL) coating (Fig. 1 C and Video 1). Over a 3-min observation period, 74% of tdT/Ly6G⁺ neutrophils exhibited multiple Ca²⁺ transients lasting for 24 ± 3 s at a rate of 1.3 ± 0.1 peaks/min (Fig. 1 D). To compare these Ca²⁺ signals with those previously reported in adherent neutrophils, we loaded neutrophils purified from non-transgenic mice with Fura-2. Fura-2 reported slower and delayed Ca²⁺ elevations whose frequency and amplitude was half those reported by Salsa6f (Fig. 1 E, green versus pink symbols). Fura-2 loading of tdT/Ly6G⁺ neutrophils reduced the amplitude and prolonged the Ca²⁺ transients recorded by Salsa6f (Fig. 1 E, blue symbols), indicating that dye loading buffers Ca²⁺ responses. The Salsa6f fluorescence outlined neutrophils polylobed nuclei, while a bright Fura-2 signal was observed in neutrophil nuclei (Fig. 1 F), indicating that only the genetic probe was confined in the cytosol. In situ calibration yielded a K_D of 160 nM for Salsa6f with a high cooperativity (Hill coefficient 3.3 versus 1.7 for Fura-2, Fig. 1 G and Fig. S1 C), similar to values reported in T cells (Dong et al., 2017). Using total internal reflection fluorescence (TIRF)

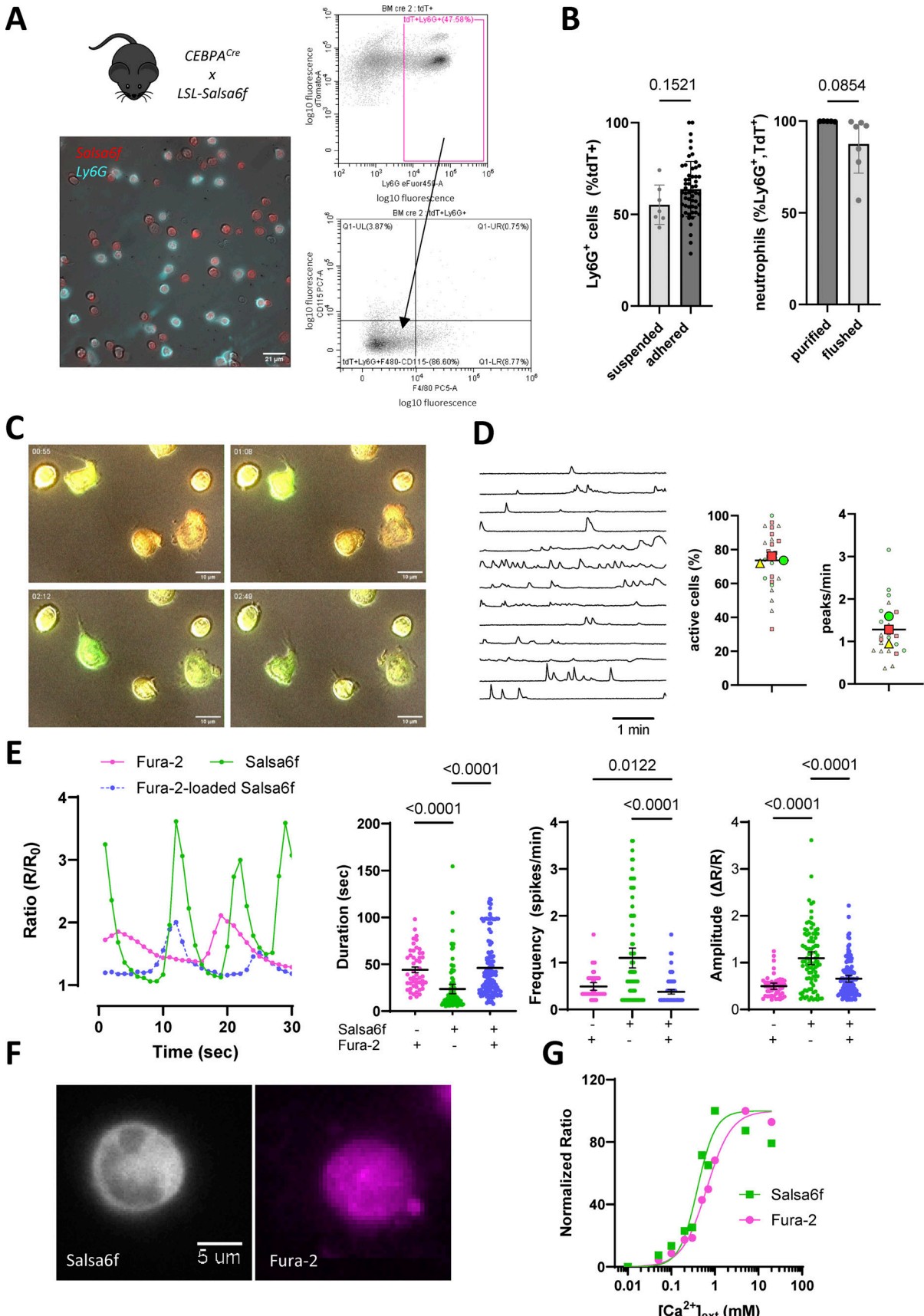

Figure 1. **Ca²⁺ elevations reported by Salsa6f in adherent murine neutrophils. (A)** Representative micrograph and flow cytometry profiles of BM cells from CEBPa-Salsa6f mice. Myeloid Salsa6f⁺ cells have a high tdT fluorescence (red) and most express the granulocyte marker Ly6G (cyan). **(B)** Proportion of

Ly6G$^+$ cells in sorted and adhered TdT$^+$ BM cells (left, $n$ = 7/54, Student's unpaired two-tailed $t$ test) and proportion of confirmed neutrophils (CD115$^-$F4/80$^-$) in TdT$^+$Ly6G$^+$ cells purified with a neutrophil kit or flushed from BM (right, $n$ = 5/7, Student's unpaired two-tailed $t$ test, cells termed "purified" and "flushed" thereafter). **(C)** Representative micrographs of the Ca$^{2+}$ fluctuations reported by Salsa6f in purified neutrophils plated on PLL. Ca$^{2+}$ elevations increase the green fluorescence of GCamP6f. See Video 1. **(D)** Representative Ca$^{2+}$ recordings of flushed murine neutrophils crawling on PLL (left) and incidence and frequency of Ca$^{2+}$ spikes. (right, $n$ = 25 recordings from three mice). **(E)** Ca$^{2+}$ fluctuations recorded by Fura-2 and Salsa6f in purified neutrophils loaded or not with Fura-2 (left) and kinetic properties of the Ca$^{2+}$ transients recorded in each condition (right, $n$ = 49/90/122 cells from three mice). **(F)** Representative micrographs of Fura-2–loaded Salsa6f$^+$-purified neutrophils, showing the cellular distribution of the TdT and Fura-2 fluorescence. **(G)** In situ calibration of Salsa6f and Fura-2 in cells equilibrated at increasing extracellular [Ca$^{2+}$] with ionomycin. $n$ = 3 mice.

---

imaging, we detected Ca$^{2+}$ waves traveling at a speed of ~30 µm/s in the TIRF plane with Salsa6f (Fig. S1 D and Video 1), Video 2 confirming the presence of actively propagating Ca$^{2+}$ signals near adhesion sites in mouse neutrophils. The high Ca$^{2+}$ sensitivity, low buffering, and cytosolic confinement of the genetic probe thus outperforms the EGTA-based dye Fura-2 and uncovers a high degree of spontaneous Ca$^{2+}$ activity in neutrophils adhered to weakly adhesive coating.

To clarify the molecular basis of neutrophil spreading-associated Ca$^{2+}$ signals, we generated mice lacking both Stim1 and Stim2 isoforms in myeloid cells (*Stim1/2$^{KO}$*), which we then bred with *Salsa6f* mice. The Ca$^{2+}$ responses evoked by Ca$^{2+}$ re-admission to cells treated with the SERCA inhibitor thapsigargin (Tg), measured with Fluo-8 or Salsa6f, were reduced by ~70% in suspended neutrophils purified from *Stim1/2$^{KO}$* mice (Fig. S1, E and F), validating the loss of functional SOCE in both animal models, while the ROS production evoked by the PKC activation with PMA, which is Ca$^{2+}$ independent, was preserved (Fig. S1 G). A similar ~70% reduction in the Ca$^{2+}$ elevations evoked by Tg and by the chemotactic peptide N-formyl-Met-Ile-Val-Ile-Leu was observed in suspended *Stim1/2$^{KO}$* neutrophils (Fig. 2 A), the reduction increasing to ~80% in adherent *Stim1/2$^{KO}$* neutrophils when measured with Salsa6f (Fig. 2 B). *Stim1/2* ablation thus severely blunts Ca$^{2+}$ responses evoked by store depletion and chemoattractants in neutrophils. We then compared the Ca$^{2+}$ signals occurring in the absence of external stimuli in neutrophils adhered to weakly adhesive coating. The Ca$^{2+}$ transients associated with neutrophil adhesion and spreading were severely impacted by *Stim1/2* ablation, the percentage of active cells, and the integrated response decreasing by 80% in *Stim1/2$^{KO}$* cells (Fig. 2 C and Video 3 and Video 4). These data show that Stim1/2 proteins sustain Ca$^{2+}$ signals associated with neutrophil spreading.

To explore the signaling pathways underlying the Stim-dependent Ca$^{2+}$ activity, we compared the effect of pharmacological inhibitors to the effect of solvent. DMSO did not alter the frequency of Ca$^{2+}$ transients or the Ca$^{2+}$ response integrated over a 3-min period (Fig. 3 A). Gadolinium, a potent Ca$^{2+}$ channel blocker, did not alter the frequency but slightly reduced the integrated response (Fig. S2 A), while GSK-7975a and CM4620, more specific Orai inhibitors, had no effect on these two parameters (Fig. 3 A and Fig. S2 A). Nonetheless, GSK-7975a decreased the Ca$^{2+}$ response evoked by Tg as efficiently as EGTA (Fig. S2, B and C), and GSK-7975a and CM4620 reduced the response evoked by Ca$^{2+}$ readmission to Tg-treated cells by 43 and 65%, respectively (Fig. S2 D), indicating that the Orai inhibitors effectively inhibited SOCE. In contrast, the PLC inhibitor U73122 immediately stopped the Ca$^{2+}$ transients (Fig. 3, A and B; and

Video 5). Among the 26 other compounds tested, only the PKC activator PMA significantly decreased the Ca$^{2+}$ activity with a delay of 40 sec that precluded statistical evaluation over the 3-min recording period (Fig. S2 A and Table S1). U73122 also abrogated the N-formyl-Met-Ile-Val-Ile-Leu–evoked response (Fig. S2 B), validating the inhibition of PLC. Several agonists that induced Ca$^{2+}$ responses accelerated the frequency of spontaneous transients, further linking this activity to PLC-coupled signaling (Table S1). These data indicate that the Ca$^{2+}$ activity of adherent neutrophils is PLC dependent and requires Stim1/2 proteins but not Ca$^{2+}$ entry, suggesting an Orai-independent function of Stim proteins.

To relate the Ca$^{2+}$ activity to adhesion-dependent cellular events, we quantified morphological changes occurring during the Ca$^{2+}$ recordings (Video 3 and Video 4), using the stable tdTomato fluorescence of WT and *Stim1/2$^{KO}$* neutrophils to outline cells. *Stim1/2$^{KO}$* neutrophils had a ~15% smaller fluorescence area and a significantly higher circularity than WT neutrophils, whose fluorescence area increased and centroid drifted during the recordings, while the area of *Stim1/2$^{KO}$* neutrophils remained constant and their centroid stationary (Fig. 4 A and Fig. S3 A). To better resolve the underlying morphological changes, we use differential interferential contrast (DIC) microscopy. Thin lamellipodia promptly appeared on the ventral side of WT neutrophils that touched the substrate, while *Stim1/2$^{KO}$* neutrophils often exhibited pseudopods that bulged and retracted (Fig. 4 B and Video 6 and Video 7). Quantitative analysis of the DIC images showed that the fraction of spread neutrophils was significantly reduced by *Stim1/2* ablation in cells adhered for ~20 min at 37°C (Fig. 4 B, graphs). Neutrophils treated with U73122 failed to spread regardless of their genotype (Fig. S3 B). We then used TIRF microscopy to quantify the formation of lamellipodia in neutrophils labeled with a PM dye. Fluorescent footprints appeared immediately in the TIRF plane as WT neutrophils landed on PLL-coated glass, reaching maximal size within 5 min, while footprints of *Stim1/2$^{KO}$* neutrophils had a reduced area 12 min after plating (Fig. 4 C and Video 8 and Video 9). Footprints forming in WT neutrophils treated with U73122 had a similarly reduced area (Fig. S3 C), indicating that *Stim1/2* ablation partially phenocopied PLC inhibition. These data indicate that *Stim1/2* ablation reduces the PLC-dependent spreading of neutrophils.

STIM proteins are reversible tethers that recruit and elongate cortical ER (cER) sheets as they interact with PM lipids and with Orai channels at ER-PM membrane contact sites (MCS) (Henry et al., 2022; Orci et al., 2009). Expecting that *Stim1/2* ablation would impact MCS formation, thereby altering Ca$^{2+}$ signals, we quantified the abundance, length, and PM proximity of cER

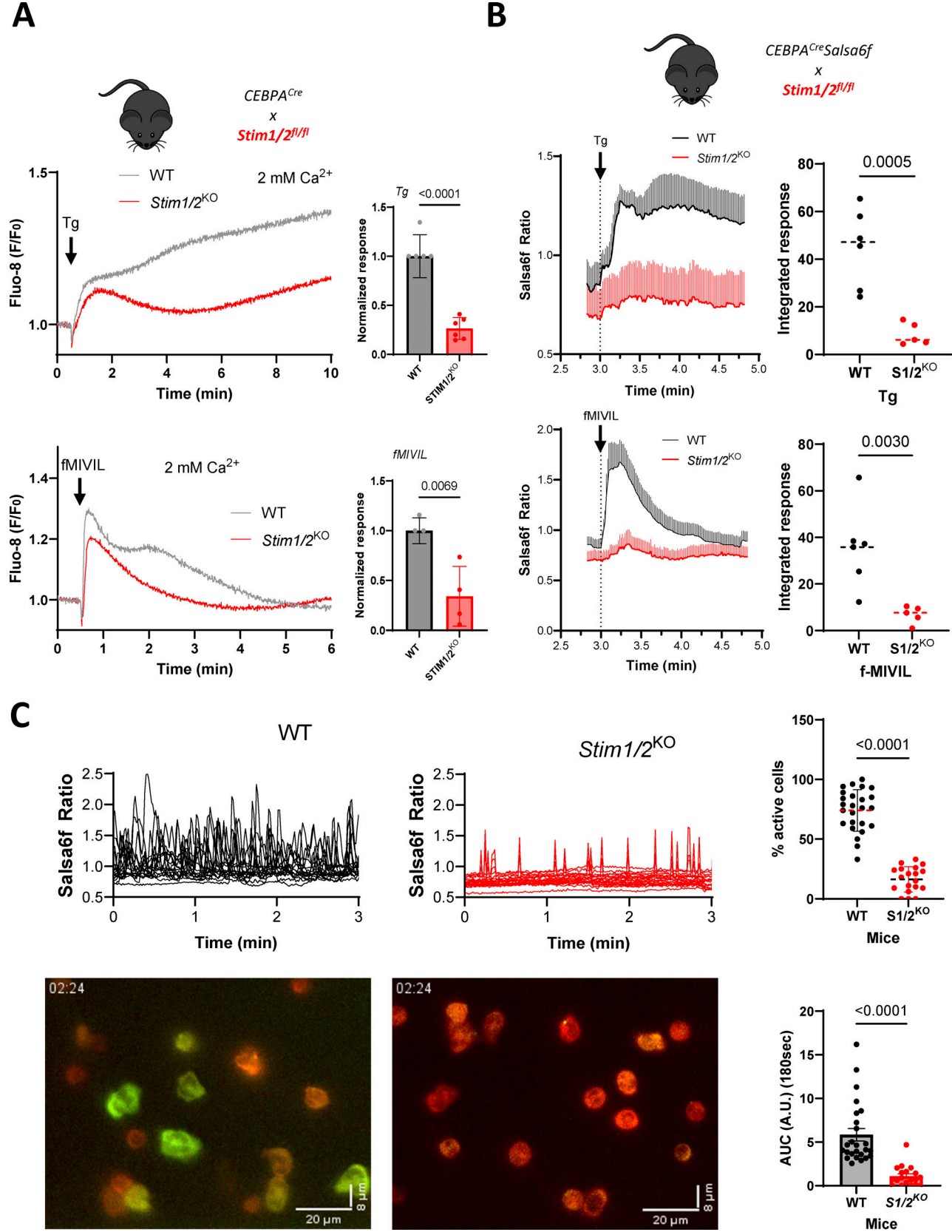

Figure 2. **Stim1/2 ablation prevents repetitive Ca²⁺ signals during neutrophils spreading. (A)** Fluo-8 recordings and quantification of Ca²⁺ responses evoked by Tg and fMIVIL in WT and *Stim1/2*^KO-purified neutrophils suspended in physiological saline. *n* = 6/5 recordings from three pairs of mice, Student's unpaired one-tailed *t* test. **(B)** Single-cell Salsa6f recordings and quantification of Ca²⁺ responses evoked by Tg and fMIVIL in flushed WT and *Stim1/2*^KO neutrophils adhered to PLL. *n* = 6/5 recordings from three pairs of mice, mean ± SEM, Student's unpaired one-tailed *t* test. **(C)** Salsa6f micrographs and

recordings of flushed cells adhered to PLL coatings (left, see Video 3 and Video 4) and incidence of $Ca^{2+}$ transients and integrated $Ca^{2+}$ responses (right, $n = 25/19$ recordings from three mice pairs, two-tailed Welch's $t$ test). fMIVIL; N-formyl-Met-Ile-Val-Ile-Leu.

structures by electron microscopy. Neutrophils purified by magnetic separation were fixed by PFA either in suspension or after adhesion to PLL, and all cER structures located <35 nm from the PM were identified visually on electron micrographs (Fig. 5 A). Outlining cellular sections indicated that WT and *Stim1/2*-deficient cells had comparable sizes and that adherent *Stim1/2*-deficient neutrophils were rounder than WT cells, in line with their spreading defect (Fig. S4 A). Contrary to our expectations, a similar proportion of cER was detected in WT versus *Stim1/2*-deficient neutrophils before or after adhesion, without noticeable differences in the lengths and ER-PM gap distances of individual cER structures (Fig. 5 B, compare gray and red symbols). Interestingly, adhesion significantly decreased the cER abundance (% cER) and lateral dimension (cER length) in WT but not *Stim1/2*-deficient neutrophils, which instead had shorter gap distances once adhered (Fig. 5 B, compare "Susp" and "Adh" conditions). Inspection of the electron micrographs indicated that WT neutrophils had adhesive membranes covered by a thick cytoskeleton that was largely devoid of contact sites (Fig. 5 A). Accordingly, a significantly lower number of MCS was detected on the adhesive membrane of WT cells compared with *Stim1/2*-deficient neutrophils, while a comparable amount was counted on the nonadhesive membranes in the two genotypes (Fig. S4 B). Morphometric analysis yielded a significantly higher proportion of cER structures of identical lengths and gap distances on adhesive membranes of *Stim1/2*-deficient neutrophils (Fig. 5 C and Fig. S4 C). We then examined the ultrastructure of adherent neutrophils treated with Tg. In this condition, WT neutrophils were larger than *Stim1/2*-deficient neutrophils, had a reduced proportion of MCS that were shorter, in closer proximity to the PM, and depleted from adhesive membranes (Fig. S4 D). These data indicate that contact sites are dynamically excluded from adhesive membranes of spreading neutrophils. Unexpectedly, *Stim1/2* ablation increased the abundance of contact sites specifically at adhesive membranes but did not alter their overall proportion or morphology.

Since contact sites were morphologically preserved in *Stim1/2*-deficient neutrophils, we compared their molecular composition by proximity ligation assays (PLAs). Cells were adhered to PLL-coated substrates, fixed, permeabilized, and stained with antibodies targeting endogenous ER and PM components. A stronger proximity signal was detected between STIM1 and Orai1 and between STIM1 and IP$_3$Rs in WT than in *Stim1/2*-deficient neutrophils (Fig. S5 A), validating the assay. We then assayed the proximity between the ER $Ca^{2+}$ release channel IP$_3$R3 and the PM $Ca^{2+}$ pump plasma membrane $Ca^{2+}$ ATPase (PMCA). The PLA signal detected between IP$_3$Rs and PMCA in WT cells was significantly reduced in *Stim1/2*-deficient neutrophils (Fig. 6 A). Histochemical analysis revealed that PMCA levels were not altered by *Stim1/2* ablation and that IP$_3$R immunoreactivity was reduced (Fig. S5 B). Similar amounts of IP$_3$R1 and IP$_3$R3 were detected by western blot in WT and *Stim1/*

*2*-deficient neutrophils (Fig. 6 B), ruling out a global decrease in IP$_3$R levels and prompting us to quantify their intracellular distribution by immunogold. A reduced total IP$_3$R1 immunoreactivity was detected in *Stim1/2*-deficient neutrophils by immunogold, the total number of gold particles, and those located <50 nm from the PM decreasing by 62% and 56%, respectively (Fig. 6 C). The mRNA levels of all three IP3Rs isoforms were comparable in the two genotypes, while mRNA levels of two membrane tethering proteins, junctate, and E-Syt2, were significantly increased in *Stim1/2*-deficient neutrophils (Fig. S6 A). These data indicate that *Stim1/2* ablation posttranslationally modifies IP$_3$R1 conformation to a less immunoreactive form and that other tethering proteins compensate the lack of Stim1/2. Altered functionality of IP$_3$Rs may further reduce the amplitude of $Ca^{2+}$ elevations and limit signal propagation. Since the $Ca^{2+}$ release activity of IP$_3$R requires a permissive cytosolic $Ca^{2+}$ concentration (Bezprozvanny et al., 1991), we then compared the resting cytosolic $Ca^{2+}$ levels ($[Ca^{2+}]_{cyt}$) of WT and *Stim1/2*-deficient neutrophils. Inactive cells were selected on Salsa6f recordings and their basal levels compared head-to-head on two different microscopy setups (Fig. 6 D and Fig. S5 C). Applying the Salsa6f calibration curves (Fig. 1 G) yielded a resting $[Ca^{2+}]_{cyt}$ of 54 nM in WT neutrophils and a significantly lower value of 42 nM in *Stim1/2*-deficient neutrophils (Fig. 6 D). These data indicate that *Stim1/2* ablation decreases the resting $[Ca^{2+}]_{cyt}$ in mouse neutrophils.

To test the reliance of *Stim1/2*-dependent functions on $Ca^{2+}$ entry, we increased the external $Ca^{2+}$ concentration to supraphysiological levels from 2 to 10 mM. Remarkably, the proportion of cells exhibiting $Ca^{2+}$ transients increased threefold within minutes after switching from physiological to $Ca^{2+}$-rich solution, and basal $Ca^{2+}$ levels were significantly increased at this point (Fig. 6 E). The $Ca^{2+}$ activity regained by *Stim1/2*-deficient cells placed in a $Ca^{2+}$-rich solution was fully inhibited by U73122, confirming its PLC dependency (Fig. 6 E). We then quantified the impact of $Ca^{2+}$ supplementation on spreading efficiency by TIRF microscopy. *Stim1/2*-deficient neutrophils developed significantly larger footprints when adhered in 10 mM $Ca^{2+}$ than in standard 2 mM saline (Fig. 6 F), while the $Ca^{2+}$ activity and spreading capacity of WT neutrophils were comparable in 2 and 10 mM $[Ca^{2+}]_{ext}$ (Fig. S5 D). $Ca^{2+}$ supplementation thus restores signaling and spreading competence in Stim1/2-deficient neutrophils.

Neutrophil spreading is an actin-driven amoeboid process and $Ca^{2+}$ elevations regulate the remodeling of the actin cytoskeleton in human neutrophils (Jaconi et al., 1991). To assess the impact of Stim1/2 deficiency on actin remodeling, we delineated in electron micrographs of adherent neutrophils all cytosolic regions lacking electron-dense structures above adhesive membranes. The basal pads overlaid on adhesive membranes were significantly smaller and thinner in Stim1/2$^{KO}$ neutrophils (Fig. 7 A and Fig. S6 B). We then used the live F-actin probe SiR-actin to dynamically record the formation of actin filaments

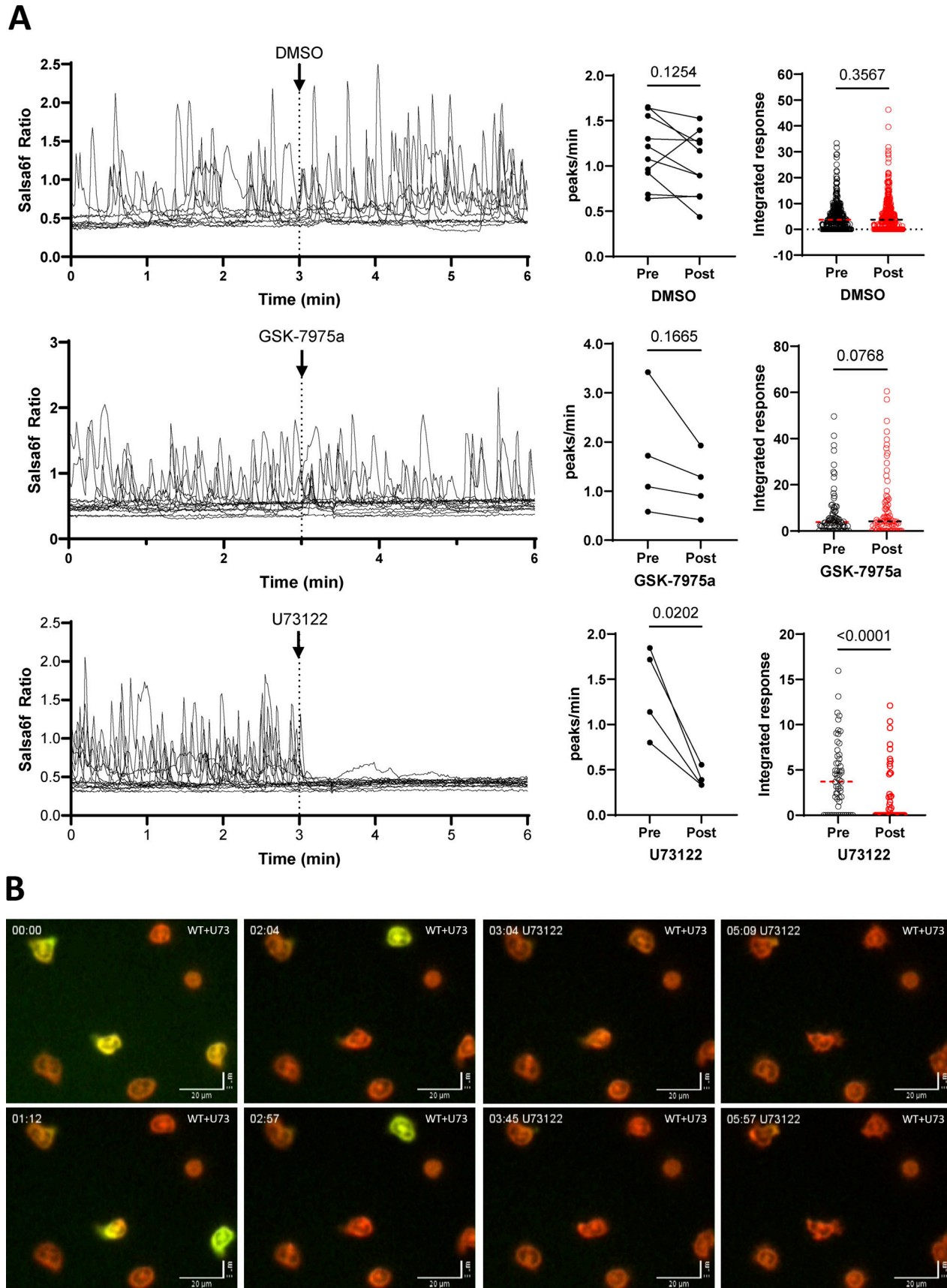

Figure 3. **Effect of Orai1 and PLC inhibition on Ca²⁺ signals during neutrophils spreading. (A)** Salsa6f recordings (left), Ca²⁺ transient frequency (middle), and integrated Ca²⁺ responses (right) of flushed neutrophils adhered to PLL and exposed to DMSO (0.1%), the Orai1 inhibitor GSK-7975a (10 μM), or the PLC

inhibitor U73122 (1 µM) in 2 mM [Ca$^{2+}$]$_{ext}$. $n$ = 436/83/56 cells in 10/4/4 recordings from four to eight mice, Student's paired two-tailed $t$ test. **(B)** Time-course micrographs of bottom recordings in A. See Video 5.

during neutrophils spreading. Transient localized increases in SiR-actin fluorescence were observed during spreading of WT neutrophils to PLL whose intensity and duration were significantly reduced in Stim1/2$^{KO}$ cells (Fig. 7 B, Fig. S6 C, Video 10, and Video 11). A reduced phalloidin staining was also detected by confocal microscopy in Stim1/2$^{KO}$ neutrophils on adhesive membranes (Fig. S6 C). These data indicate that actin dynamic is impaired in Stim1/2-deficient neutrophils.

To assess the physiological role of Stim1/2 during the acute inflammatory response in vivo, we analyzed extravasation of *Stim1/2$^{KO}$* neutrophils in inflamed murine cremaster muscles. Male mice were injected i.p. with 500 ng of TNF for 2 h, euthanized, and their cremaster muscles were harvested and stained using anti-CD31 and anti–GR-1 antibodies to label blood vessels and granulocytes, respectively. Interestingly, our results revealed a significant increase in extravasation of *Stim1/2$^{KO}$* neutrophils into the inflamed tissue compared with WT animals (Fig. 8). Together, these data suggest that Stim 1/2 are crucial in regulating neutrophil migration from blood into the tissue under acute inflammatory conditions in vivo.

## Discussion

Our study identifies Stim proteins as critical regulators of neutrophil spreading. *Stim1/2*-deficient murine neutrophils lacked PLC-dependent Ca$^{2+}$ elevations during adhesion and poorly formed lamellipodia when plated on solid substrates. *Stim1/2* ablation aborted SOCE and decreased Ca$^{2+}$ elevations evoked by chemokines or bacterial peptides, consistent with earlier reports in mouse neutrophils (Nunes et al., 2012; Sogkas et al., 2015; Zhang et al., 2014). Using myeloid-targeted Salsa6f, we show that *Stim1/2* deficiency also reduces basal Ca$^{2+}$ levels and prevents cytosolic Ca$^{2+}$ elevations in adherent mouse neutrophils. The reduced buffering and clean cytosolic confinement of the genetic probe facilitated the quantification of basal levels and the detection of low-amplitude cytosolic Ca$^{2+}$ elevations. Surprisingly, the Ca$^{2+}$ flickers were insensitive to Orai1 inhibition, indicating that Stim1/2 sustain Ca$^{2+}$ elevations independently of Orai1 during the engagement of adhesion receptors. This is congruent with earlier reports showing that Orai1 deficiencies do not phenocopy *Stim1* or *Stim1/2* deficiencies in mouse neutrophils (Clemens et al., 2017; Dixit et al., 2011; Schaff et al., 2010; Sogkas et al., 2015; Zhang et al., 2014).

STIM2 was identified in a siRNA screen as a regulator of basal cytosolic and ER Ca$^{2+}$ levels (Brandman et al., 2007) and *Stim2* deficiency might cause the reduced basal Ca$^{2+}$ levels of *Stim1/2*-deficient neutrophils. The reduced [Ca$^{2+}$]$_{cyt}$, in turn, might account for the loss of PLC-dependent Ca$^{2+}$ flickers, as the activity of IP$_3$Rs requires a permissive cytosolic Ca$^{2+}$ concentration (Arige et al., 2022). The rapid recovery of Ca$^{2+}$ flickers in *Stim1/2*-deficient neutrophils exposed to Ca$^{2+}$-rich (10 mM) solutions strongly support this hypothesis and indicate that *Stim1/2*

deficiency restricts Ca$^{2+}$ influx via GSK-7975a and CM4620 (zegocractin)-insensitive Ca$^{2+}$ channels.

STIMs are multifunction proteins acting as ER sensors via their luminal helix–loop–helix (EF-hand) motif, as ER-PM tethering proteins via their lipid-binding domains, and as intracellular ligands for Ca$^{2+}$ channels via their channel activating and polybasic domains. The predominant and best studied function of STIMs is to generate Ca$^{2+}$ fluxes by interacting with PM channels, but their ER-PM tethering function is equally important to allow the fluxes of ions and the exchange of lipids between membranes. This motivated us to assess how *Stim1/2* deficiency impacts ER-PM tethering by performing an ultrastructural analysis of adherent and nonadherent neutrophils. We made several unexpected observations by electron microscopy. First, the cER abundance, shape, and proximity were not impacted by *Stim1/2* ablation, despite a near complete loss of SOCE. In cell lines, *Stim1* ablation reduces the proportion of cER threefold, while STIM1 overexpression has the opposite effect, and store depletion both increases the amount of cER and elongates cER sheets (Henry et al., 2022; Orci et al., 2009; Saüc et al., 2015). We were therefore expecting to find a reduced cER proportion and shorter sheets in Stim1/2-deficient neutrophils. Instead, a similar amount of morphologically indistinguishable cER sheets was quantified in both genotypes, regardless of the adhesion state of neutrophils. The higher mRNA levels of junctate and E-Syt2 in Stim1/2-deficient cells suggest that these other tethering proteins are compensating for the loss of *Stim1/2* in neutrophils. Second, cER structures were virtually absent from membranes interacting with the substrate in WT neutrophils (adhesive membranes) but persisted in adhesive membranes of Stim1/2-deficient neutrophils, which had a thinner actin cytoskeleton and impaired actin dynamics during spreading.

We offer two non-mutually exclusive mechanistic explanations for these differences. (1) The absence of Ca$^{2+}$ signals at contact sites forming in Stim1/2-deficient neutrophils might prevent efficient actin remodeling, as Ca$^{2+}$ elevations control both actin polymerization and depolymerization by driving the activity of Ca$^{2+}$-dependent proteins such as calpain and gelsolin (Larson et al., 2005; Lokuta et al., 2003). *Stim1/2*-deficient neutrophils regained spreading competence in Ca$^{2+}$-rich solutions, supporting this Ca$^{2+}$-centric hypothesis. (2) The Stim1/2-deficient contact sites might be too tightly attached to the PM, hindering the formation of the actin cytoskeleton during neutrophil adhesion. The binding of STIM proteins to the PM is reversible and controlled by changes in ER and cytosolic (Ca$^{2+}$) and in PM phosphoinositides. No other tethering protein has such a regulation, ideally suited to promote the rapid formation and subsequent detachment of contact sites during PLC-mediated signaling. Junctate is an ER Ca$^{2+}$-sensing protein that is co-recruited with STIM1 to ER-PM contact sites upon store depletion (Srikanth et al., 2012), while E-Syt2 interacts with E-Syt1, which mediates Ca$^{2+}$-dependent ER-PM tethering and

**A**

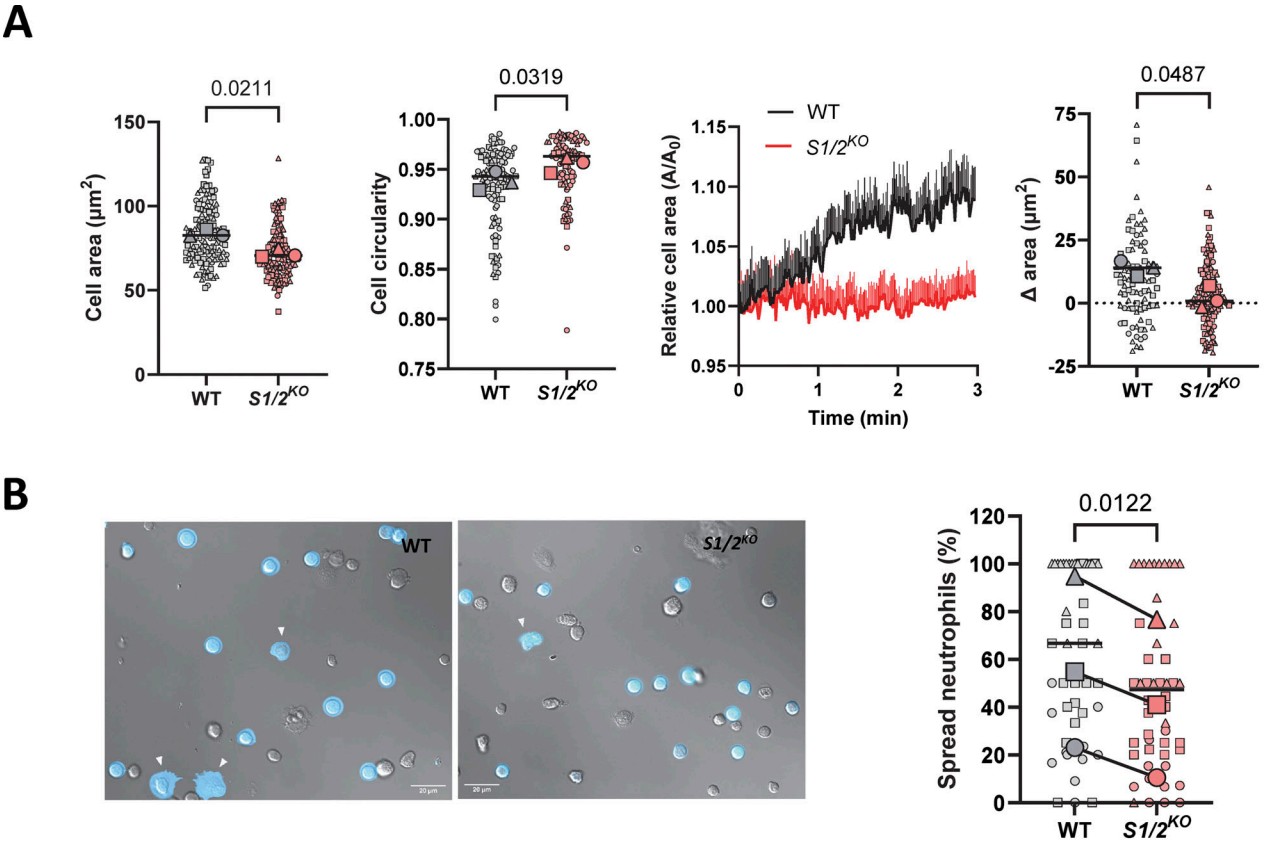

**B**

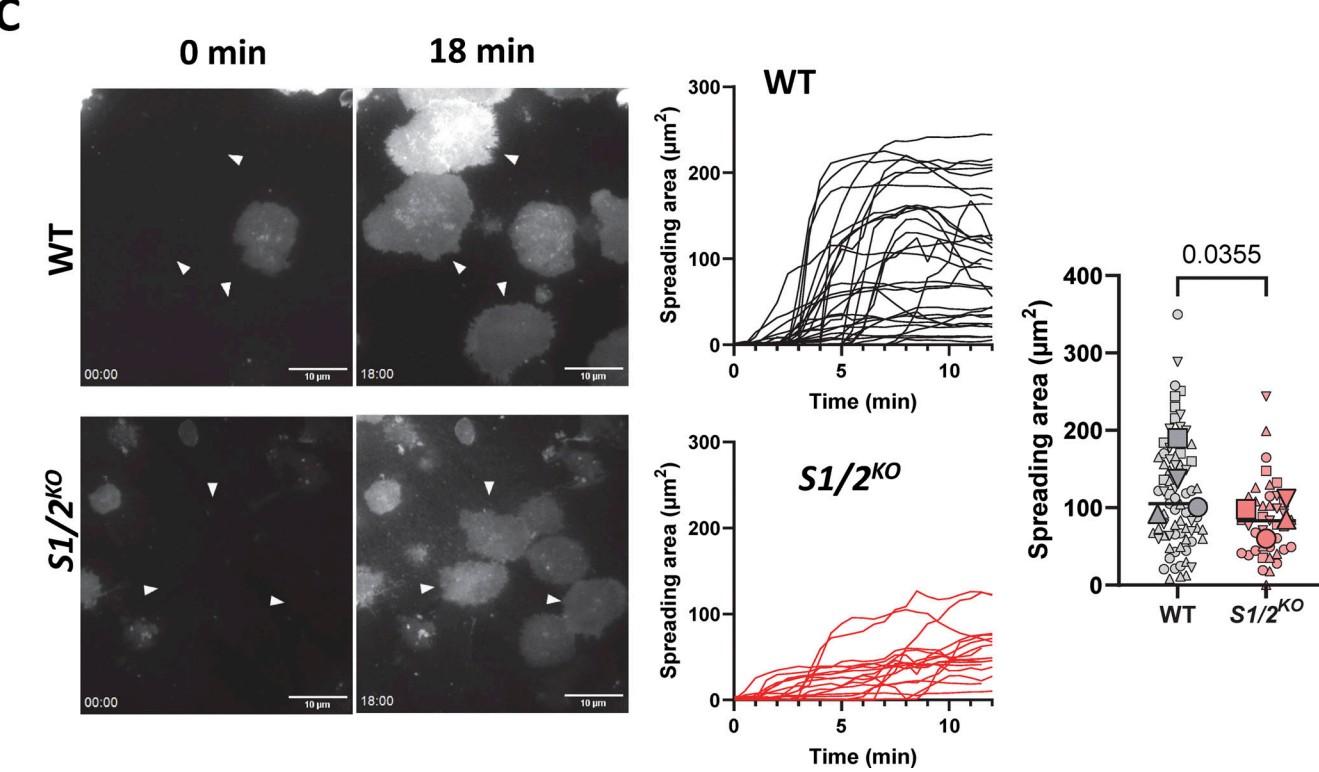

**C**

Figure 4. **Effect of Stim1/2 ablation on the spreading of neutrophils. (A)** Average tdTomato fluorescence area and circularity of WT and *Stim1/2$^{KO}$*-flushed neutrophils plated on PLL during the 3-min recordings in Fig. 2 and relative changes in the fluorescence area of these cells (*n* = 127/134 cells in 7/7 recordings

from three WT/*Stim1/2^KO* mice pairs, Student's paired one-tailed *t* test on mice pairs. Small and large symbols show data from individual cells and mouse average, lines show median cellular values). **(B)** DIC micrographs (left) and fraction of spread WT and *Stim1/2^KO*-flushed neutrophils plated on PLL for ∼20 min at 37°C and at RT (right, n = 37/39 and 11/11 recordings from 3 WT/*Stim1/2^KO* mice pairs, Mann–Whitney U test. Student's paired two-tailed *t* test on mice pairs). See Video 6 and Video 7. **(C)** TIRF micrographs of flushed neutrophils labeled with the PM dye CellMask adhering to PLL-coated glass (left, Video 8 and Video 9) and change in the size of neutrophil footprints after PLL contact (right, n = 25/25 graphed and 75/41 analyzed cells from four WT/*Stim1/2^KO* mice pairs, two-tailed Welch's *t* test on cell values).

gap shortening (Fernández-Busnadiego et al., 2015; Giordano et al., 2013). Junctate recruits ER $Ca^{2+}$ stores to phagosomes in *Stim1/2*-deficient mouse fibroblasts (Guido et al., 2015), and complementation by junctate and E-Syt2 likely accounts for the persistence of contact sites in Stim1/2-deficient cells and for their reduced gap distance in conditions promoting $Ca^{2+}$ elevations.

The altered functionality of $IP_3Rs$ at ER-PM contact sites has important functional consequences for $Ca^{2+}$ signaling. $IP_3Rs$ on PM-bound cER sheets are in a prime location to respond to PLC-coupled agonists, as the ER-PM cleft restricts the diffusion of $IP_3$ molecules generated by the local hydrolysis of $PI(4,5)P_2$ and lipid exchange at ER-PM contact sites mitigates phosphoinositide consumption by PLC (Chang and Liou, 2015; Kim et al., 2016). Previous studies indicated that only receptors located at the rim of ER-PM contact sites are in a "licensed" state able to release $Ca^{2+}$ (Thillaiappan et al., 2017). These licensed receptors might further enable $Ca^{2+}$ signal propagation by facilitated diffusion within the ER lumen to rapidly reach distant effector targets, a $Ca^{2+}$ signaling modality termed "$Ca^{2+}$ tunneling" (Courjaret et al., 2018, 2025; Courjaret and Machaca, 2014). In this model, $Ca^{2+}$, entering via SOCE or other channels at ER-PM contact sites, is rapidly pumped into the ER lumen by adjacent SERCAs and released via $IP_3R$ at distal sites following diffusion in the ER lumen. The ability of Stim proteins to act as labile tethers and to maintain $IP_3Rs$ functionality as neutrophils adhere and spread therefore influence their signaling competence in multiple ways.

While the *Stim1/2*-deficient neutrophils spread poorly in vitro, they emigrated better into tissue than control cells in an in vivo mouse model of acute inflammation. This is at odds with previous studies reporting either normal or reduced recruitment of *Stim1* or *Stim1/2*-deficient neutrophils to sites of inflammation. Normal recruitment was observed in models of established peritoneal and subcutaneous inflammation (Clemens et al., 2017; Zhang et al., 2014) and reduced recruitment in acute lung infection (Yang et al., 2023) and imiquimod-induced psoriasis (Steinckwich et al., 2015). These discrepancies were attributed to differences between acute versus established models of inflammation and to paracrine amplification of neutrophil recruitment (Yang et al., 2023). Our findings indicate that differences in the extracellular $Ca^{2+}$ concentration could also contribute to the variability of *Stim1/2*-dependent phenotypes in vivo. The extracellular $Ca^{2+}$ concentration in the blood and interstitial fluid is around 1.2 mM but decreases to 0.2–0.4 mM in the mammalian epidermis (Elias et al., 2002). These low external $Ca^{2+}$ concentrations might exacerbate the reliance of neutrophils motility on Stim1/2, accounting for the reduced recruitment of *Stim1/2*-deficient neutrophils in the inflamed skin.

We can only speculate as to how the reduced spreading of *Stim1/2*-deficient neutrophils might enhance their extravasation. Our adhesion and spreading assays were performed under static conditions, while neutrophils trans-endothelial migration is a multistep process occurring under flow conditions (Ley et al., 2007; Zarbock and Ley, 2009). STIM proteins may act differently on specific β2 integrins, in particular LFA-1 and Mac-1, sequentially engaged during trans-endothelial migration. LFA-1 plays a crucial role in the initiation of firm adhesion while Mac-1 is important for the subsequent process of intraluminal crawling under flow. In this scenario, the adhesion defect of *Stim1/2*-deficient neutrophils observed in vitro could be primarily due to impaired LFA-1 function. The increased extravasation in vivo, on the other hand, could be a consequence of increased Mac-1–dependent crawling, which could enhance trans-endothelial migration by allowing neutrophils to find optimal emigration sites more efficiently. This hypothesis is consistent with the findings of (Phillipson et al., 2006) and could be experimentally tested by measuring the impact of *Stim1/2* deficiency on the kinetics of β2-integrin–mediated processes during each step of neutrophil extravasation.

In summary, we show that *Stim1/2* ablation does not prevent the formation of ER-PM MCSs but deprives cER sheets from functional $IP_3Rs$ and reduces basal $Ca^{2+}$ levels, impairing adhesion-driven $Ca^{2+}$ signals, actin dynamics, and neutrophil spreading. The tethering proteins that compensate for the lack of Stim1/2 generate persistent $IP_3Rs$-deficient MCSs that impede actin remodeling during neutrophil spreading. Stim proteins thus maintain signaling competence in spreading neutrophils by controlling basal $Ca^{2+}$ levels and the dynamic delivery and retrieval of $IP_3Rs$-containing cER sheets at sites of receptor engagement.

## Materials and methods

### Antibodies and reagents

The following reagents were used in this manuscript; Thapsigargin (67526-95-8; Sigma-Aldrich), Fura-2-AM (FP-42776A; Interchim), Deep Red CellMask (C10046; Thermo Fisher Scientific), and Fluo-8-AM (21080; AAT Bioquest). Mouse anti-IP$_3$R3 (610313; Biosciences; Monoclonal [2/IP3R-3]), rabbit anti-IP$_3$R1 (ab5804; abcam; polyclonal), rabbit anti-Stim1 (S6072; Sigma-Aldrich; polyclonal), mouse anti-Orai1 (ab175040; Abcam; 266.1), mouse anti-PMCA (ab2825; Abcam; 5F10), and mouse anti-vinculin (66305-1; Proteintech). Goat anti-mouse Alexa Fluor 488 (A-11029; Thermo Fisher Scientific) and goat anti-rabbit Alexa Fluor 555 (A-21428; Thermo Fisher Scientific) or goat anti-mouse and anti-rabbit coupled to horseradish peroxidase (Bio-Rad) were used as secondary antibodies. $Ca^{2+}$ recordings

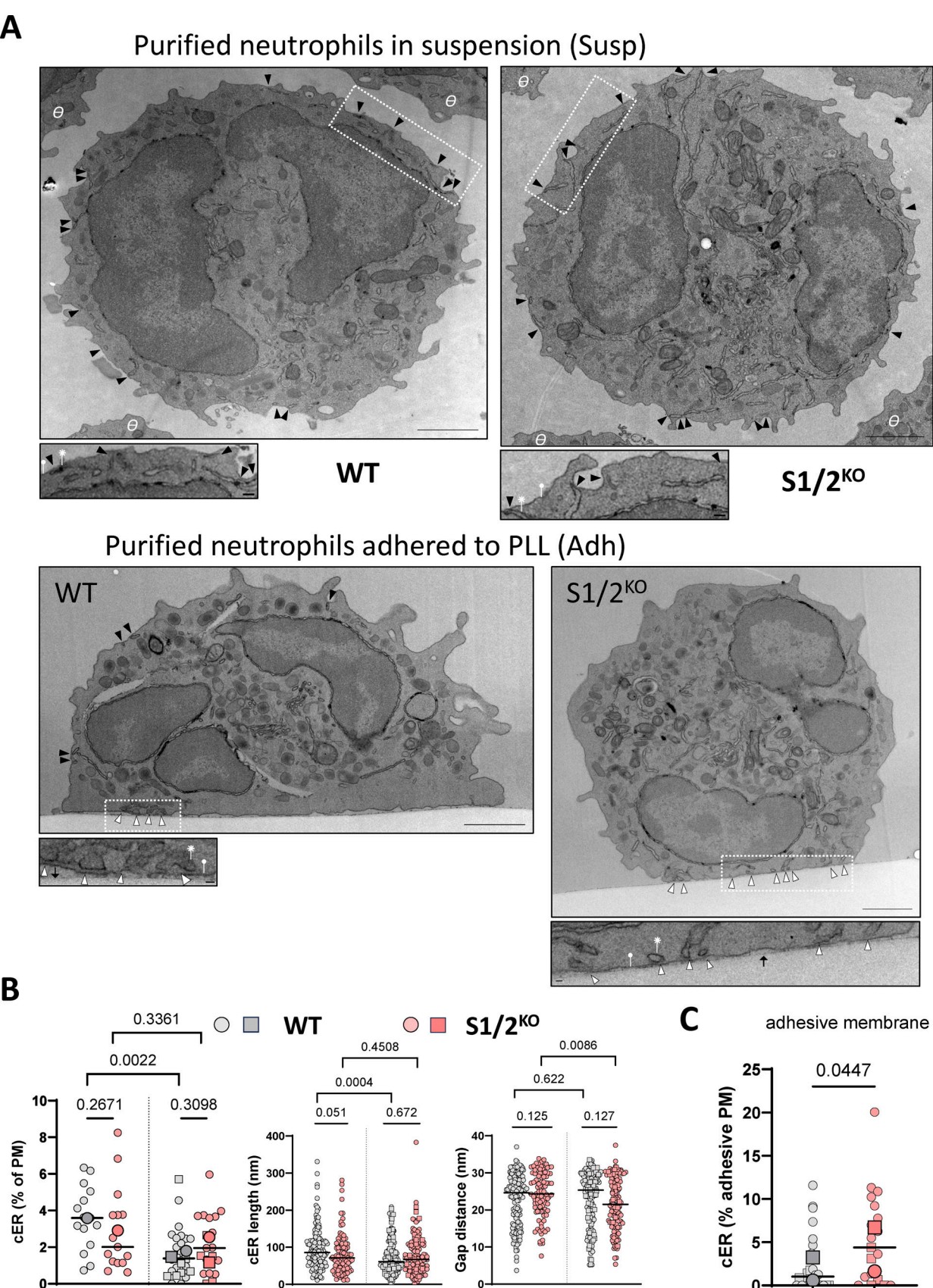

Figure 5. **Effect of Stim1/2 ablation on the ultrastructure of mouse neutrophils. (A)** Representative electron micrographs of suspended (top) and adherent (bottom) neutrophils purified from WT (left) and *Stim1/2KO* (right) mice. Arrowheads indicate ER structures located <35 nm from the PM at unsettled

(black) or adherent (white) membranes. ϴ: adjacent cell. Scale bar: 1 µm. Insets: Higher magnification micrographs showing cER sheets (asterisks) apposed to the PM (closed circle). Black arrow denotes the dish bottom. Scale bar: 40 nm. Micrograph of adherent *Stim1/2*[KO] neutrophil is replicated in Fig. 7 A. **(B)** Percentage of PM decorated by cER and quantification of cER length and ER-PM gap distance in suspended (Susp) and adherent (Adh) WT and *Stim1/2*[KO] neutrophils. *n* = 15/15 Susp, 26/19 Adh cells with 148/95 cER sheets and 150/124 cER sheets from two mice pairs, Mann–Whitney U test. **(C)** Percentage of adhesive membrane with cER. *n* = 26/19 cells. Lines show median values, larger symbols mouse average.

and live imaging experiments were conducted in physiological buffer containing 140 mM NaCl, 5 mM KCl, 1 nM MgCl$_2$, 2 mM CaCl2, 20 mM Hepes, and 10 mM glucose, pH 7.4, with NaOH.

## Mouse models

Mice deficient for Stim1 and Stim2 in the myeloid lineage were generated by crossing Stim1$^{fl/fl}$Stim2$^{fl/fl}$ animals in the C57BL/6 background (Oh-Hora et al., 2008) with mice expressing Cre recombinase under the endogenous *Cebpa* promoter (Cebpa$^{cre/+}$ mice) (Wölfler et al., 2010). Mice expressing the Ca$^{2+}$ reporter Salsa6f in the myeloid lineage were generated by crossing Cebpa$^{cre/+}$ mice with mice bearing two LSL-Salsa6f alleles inserted in the Rosa26 locus (Dong et al., 2017). Mice were bred in specific opportunist pathogen-free conditions at Charles River Laboratories and housed in the specific pathogen–free facility at the University of Geneva, Geneva, Switzerland. All animal manipulations were performed in accordance to the guidelines approved by the animal research committee at the University of Geneva. Genotyping was performed from ear biopsies as described in Nunes et al. (2012).

## Neutrophil isolation

BM cells were obtained from mouse femurs and tibias as previously described (Nunes et al., 2012). Flushed BM cells were kept on ice and used on the same day for imaging with post hoc neutrophil identification or for isolation of neutrophils by negative selection with a purification kit (130-097-658; Miltenyi Biotec).

## Flow cytometry

$1 \times 10^5$ BM or isolated neutrophils were labeled with antibodies against surface markers in a 96-well U-plate for 30–60 min at 4°C. Viability was assessed by co-incubation with LIVE/DEAD Fixable Yellow Dead Cell Stain at a final concentration of 1 µl/1 × $10^6$. Cells were washed and pelleted at 1,200 rpm for 5 min and fixed with 2% PFA. Cells were analyzed with CytExpert. For Ca$^{2+}$ recordings, cells were used directly (for Salsa6f) or loaded with Fluo-8 (2 µM for 30 min, RT). Flux was adjusted to 1 µl/s and 2,000 cells/s on a BDFSR Fortessa. $5 \times 10^5$ cells in 510 µl of Ca$^{2+}$-free solution (1 mM EGTA) were sequentially exposed to Tg (10 µM) and CaCl$_2$ (final concentration 2.5 mM), and the average median fluorescence was recorded every 10 s with Flowjo.

## Calcium recordings

Ca$^{2+}$ imaging was performed as described previously (Henry et al., 2022). 20–100 × $10^3$ BM or purified neutrophils were plated on a small drop in the center of a 0.001% PLL-coated 25-mm glass coverslip in microscope chambers and left to adhere for 3 min before filling the chamber with physiological buffer. Cells were imaged at 40× magnification on a Nikon Eclipse Ti microscope (Nikon Instruments) equipped with a Lambda DG4

illumination system (Sutter Instrument) and a 16-bit CMOS camera (pco.Edge sCMOS; VisitronSystems). Ca$^{2+}$ activity was recorded with Salsa6f at 490/525 nm$_{ex/em}$ and 572/630nm$_{ex/em}$ and concomitantly or separately at 340/510 nm$_{ex/em}$ and 380/510 nm$_{ex/em}$. In cells loaded with 2 µM Fura-2-AM for 20 min, using filter sets from Chroma Technology. Ca$^{2+}$ transients were identified as peaks exceeding 40% of the baseline R/R$_0$ value. In situ calibration was performed as in (Dong et al., 2017), using Fura-2–loaded isolated Salsa6f$^+$ neutrophils treated with 2 µM ionomycin in Ca$^{2+}$-free solutions and equilibrated with increasing Ca$^{2+}$ concentrations. Post hoc identification of neutrophils in BM cells was done by surface staining with eFluor 450–tagged anti-Ly6G monoclonal antibody (1A8; eBioscience) imaged at 402/460 nm$_{ex/em}$. Ca$^{2+}$ recordings of isolated neutrophils in suspension were performed on a Hamamatsu FDSS µCELL. Cells were loaded with 4 µM Fluo-8-AM for 40 min, washed, and kept in 20 mM Hepes-supplemented HBSS. 20–30 × $10^3$ cells were dispensed in 384-well plates and 5 µl of the indicated compounds added after 30 s of recordings at 490/525 nm$_{ex/em}$ at a rate of 0.5 s for 15 min.

## Western blotting

Isolated mouse neutrophils were lysed with RipA buffer, and the collected proteins quantified with Bradford reagent (Sigma-Aldrich). 20 µg (for IP$_3$R1) or 40 µg (for IP$_3$R3) proteins were loaded into 3–8% Tris-acetate polyacrylamide gels, separated by SDS-PAGE, transferred to polyvinylidene difluoride membranes, and revealed with primary rabbit and mouse antibodies and goat secondary antibodies coupled to horseradish peroxidase. Chemiluminescence was measured on a Fusion FX imaging device (Vilber), and data expressed as ratio of IP$_3$R over vinculin immunoreactivity.

## Immunolabeling and PLA assays

For immunostaining, 60 × $10^3$ purified cells were adhered for 10 min on PLL, fixed in 4% paraformaldehyde, permeabilized with 0.1% NP40, and blocked in 0.5% BSA/3.5% FCS. Cells were stained overnight at 4°C with primary antibodies and 1 h at RT with secondary antibodies diluted in 0.5% BSA before washing and mounting in SlowFade/10 µg/ml Hoeschst. Confocal images of the bottom and middle planes of cells were acquired with LSM800 Airyscan at 60× magnification. For PLA, the Duolink In Situ Red Starter Kit Mouse/Rabbit (DUO92101; Sigma-Aldrich) was used to reveal proximity between targets after overnight incubation with primary antibody pairs. Cells were incubated with anti-mouse MINUS and anti-rabbit PLUS probes for 1 h at 37°C before ligation for 1 h at 37°C and amplification for 100 min at 37°C. Fluorescence of nuclei and of proximity pairs were measured using the LSM800 Airyscan at 40× magnification using Hoechst 33342 (405/461 nm$_{ex/em}$) and Texas Red (594/624

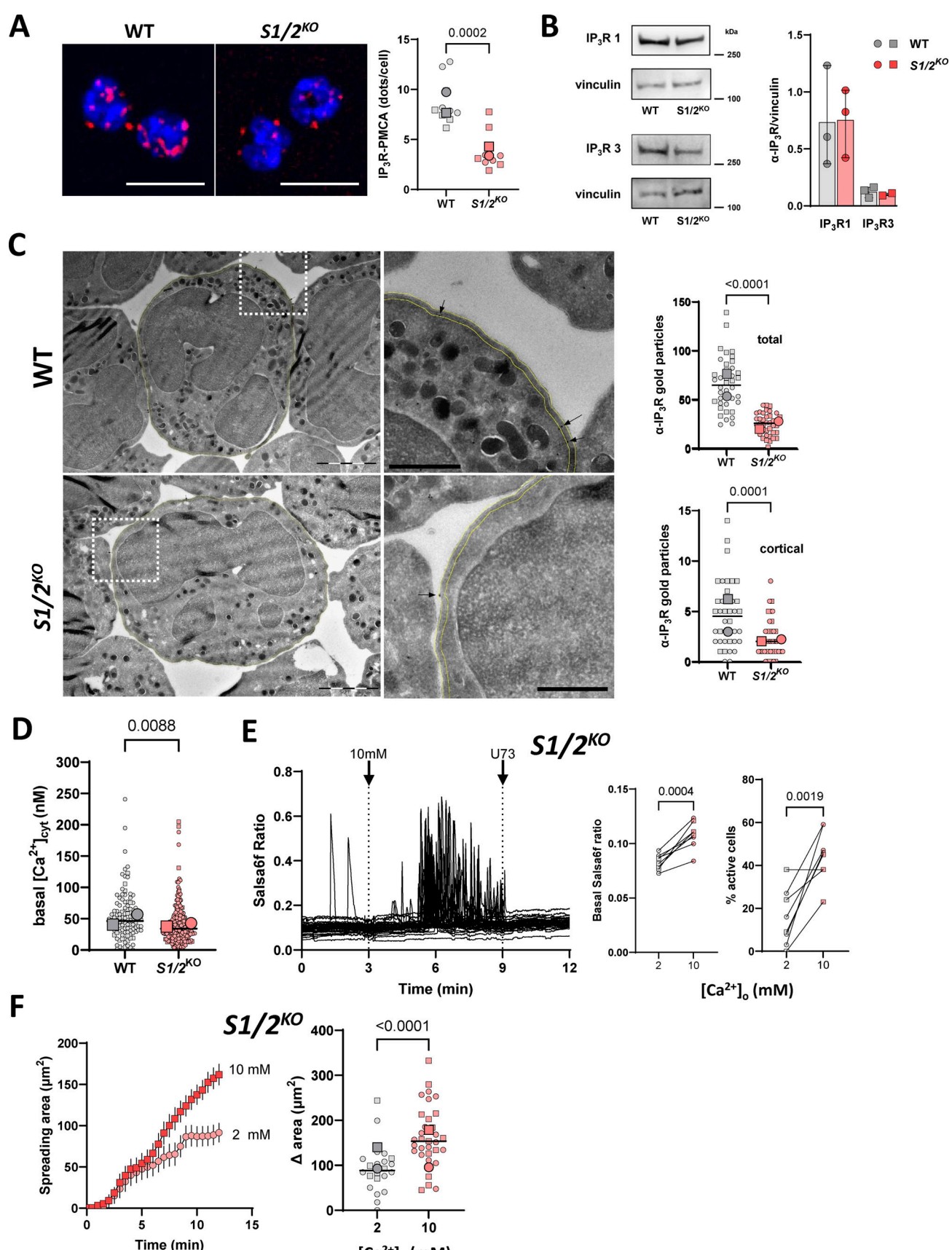

Figure 6. **Effect of Stim1/2 ablation on IP₃R expression and basal [Ca²⁺]cyt levels and effect of Ca²⁺ supplementation on Ca²⁺ activity and spreading of Stim1/2-deficient neutrophils. (A)** Representative z-stack maximal intensity projection micrographs of PLAs between IP₃R3 and the PM Ca²⁺ pump PMCA

in WT and *Stim1/2*-deficient purified neutrophils (left, red dots on the confocal micrographs, nuclei stained blue with DAPI, scale bars: 10 μm) and quantification of the interactions (right, $n$ = 10/10 micrographs with >10 cells for each condition from two mice pairs, Mann–Whitney U test). **(B)** Western blot showing IP$_3$R1 protein levels in WT and *Stim1/2*-deficient purified neutrophils. **(C)** Representative immuno-electron micrographs (left) of WT and *Stim1/2*-deficient purified neutrophils exposed to an anti-IP$_3$R1 antibody coupled to 10-nm-diameter gold particles (arrows) and quantification of the number of gold particles in the cytosol and in regions <50 nm from the PM in WT and *Stim1/2$^{KO}$* neutrophils (right, $n$ = 20/20 cells from one pair of mice, exact two-tailed Mann–Whitney U test. Scale bar: 500 nm). **(D)** Resting cytosolic Ca$^{2+}$ concentration of WT and *Stim1/2*-deficient flushed neutrophils, measured with Salsa6f in non-spiking cells. Salsa6f ratios were converted to [Ca$^{2+}$]$_{cyt}$ using the calibration curve of Fig. 1 G. $n$ = 108/230 cells from two pairs of mice, unpaired two-tailed $t$ test on cell values, lines are median values, larger symbols mouse average. **(E)** Salsa6f recordings (left), basal Ca$^{2+}$ levels (middle), and proportion of flushed *Stim1/2$^{KO}$* neutrophils exhibiting Ca$^{2+}$ transients before and after switching from 2 to 10 mM [Ca$^{2+}$]$_{ext}$ (right, $n$ = 151 cells in eight recordings from two pairs of mice, Student's paired two-tailed $t$ test). **(F)** Change in the size of CellMask TIRF footprints of flushed *Stim1/2$^{KO}$* neutrophils adhering to PLL-coated glass in 2 and 10 mM [Ca$^{2+}$]$_{ext}$ (left) and quantification of the spreading area (right, $n$ = 22/33 cells in 4/5 recordings from 2 WT/*Stim1/2$^{KO}$* mice pairs, exact two-tailed Mann–Whitney U test on cell values). Source data are available for this figure: SourceData F6.

nm$_{ex/em}$) channels. Confocal z-stacks of 0.5 μm were acquired and a maximal intensity Z-projection generated. A script was written to count nuclei and fluorescent dots based on particle size and threshold. The number of dots per cell was assessed by particle analysis with Fiji for nuclei (800-infinity pixel display) and duolink dots (2–15 pixel display). Only the dots in the vicinity of nuclei were counted.

## Morphometric analysis and spreading assays

The tdTomato signal was used for morphometric analysis and tracking of neutrophils identified by post hoc Ly6G staining on Salsa6f recordings using scrips available on GitHub. For DIC microscopy, 60 × 10$^3$ cells were adhered for 3 min and random fields acquired at fixed time points to evaluate the spreading of neutrophils identified by post hoc Ly6G staining. DIC images were acquired every 2 sec on a Zeiss Axio Observer Z1 microscope at 63× magnification. Spreading kinetics was measured on a Nikon TIRF microscope at a magnification of 100×. 10$^5$ BM cells were incubated for 15 min at 37°C with 1:10,000 CellMASK Deep Red (C10046; Thermo Fisher Scientific) in GlutaMAX RPMI 1640 medium (61870036; Thermo Fisher Scientific), washed, resuspended in 2 mM Ca$^{2+}$, and added to PLL-coated coverslips inserted in a chamber inside the TIRF microscope. A few cells were plated to find the TIRF plane using the CellMASK 640 signal. After 1 min, the bulk of the cell suspension was added, and lamellipodia appearing in the TIRF plane were imaged. Images were acquired every 30 s for 12 min. Neutrophils were identified by post hoc Ly6G staining. Changes in spreading area were analyzed using Image J/Fijy.

## Transmission electron microscopy

150 × 10$^3$ cells were either adhered on PLL-coated 12-mm glass coverslips and treated or not with Tg (1 μM for 15 min) or left in suspension before pelleting in Eppendorf and processed as in Henry et al. (2022). Briefly, cells were fixed with 2.5% glutaraldehyde, following a staining with uranyl acetate, and then postfixed with osmium tetroxide. Samples were embedded in Epon resin and then sectioned at 30 nm. Samples were imaged on a Technai 20 transmission electron microscope (FEI) at 92,000 times magnification. For each cell identified on the grid every ER structure located within 30 nm of the PM was imaged. The cell periphery was outlined to calculate the proportion of PM coverage. The cER length and the ER-PM gap distance were determined on EM images using ImageJ/Fiji.

## F-actin live imaging

Primary mouse neutrophils were loaded for 30 min at 37°C with 1 μM SiR-actin (SC001 Spirochrome) in presence of 10 μM verapamil to block efflux pumps according to the manufacturer's instructions. Actin dynamics were then imaged using a Nikon ECLIPSE Ti2-E inverted widefield fluorescence microscope equipped with a Lumencor Spectra III LED light engine at 60× magnification. Cells were gently dropped on a PLL-coated coverslip in an imaging chamber already containing 500 μl of physiological buffer. Images were acquired every 0.2 s at λ$_{ex/em}$ 652/674 nm, and changes in SiR-actin fluorescence were measured for 15 min as cells touched the coverslip.

## Induction of inflammation in vivo and whole mount immunostaining of mouse cremaster muscles

To induce inflammation, male mice were injected i.p. with recombinant murine TNFα (500 ng in 200 μl PBS per mouse, # 300-01A; PeproTech). 2 h after injection, the cremaster muscles were surgically removed. For whole mount immunostaining, extracted muscles were then fixed in formalin for 20 min, blocked, and permeabilized in PBS supplemented with 20% normal goat serum and 0.5 % Triton X-100 for 2 h. For immunostaining of blood vessels and neutrophils, muscles were incubated with fluorescently labeled primary antibodies rat anti-mouse CD31-AF647 (clone MEC13.3; Biolegend) and rat anti-mouse GR1-FITC (clone RB6-8C5; Biolegend) at RT overnight. Animal studies were performed according to the protocol GE-154 approved by the University of Geneva animal research committee.

## Imaging and analysis of neutrophil extravasation by confocal fluorescence microscopy

Images were acquired using an Axio Examiner Z1 spinning disk confocal microscope (Zeiss) equipped with 488- and 640-nm laser sources and a 20× objective using Slidebook 6.0.12 software (Intelligent Imaging Innovations). Multiple 3D fields of view (z-stacks) were captured using the software's automatic scanning mode. Using maximum projected z-stacks, the quantification of neutrophil extravasation was performed by counting all interstitial GR1$^+$ cells within a distance of 75 μm from postcapillary venules with an average diameter >30 μm. Extravasation was then calculated as extravascular cell counts per area analyzed using ImageJ (version 1.54f; FIJI).

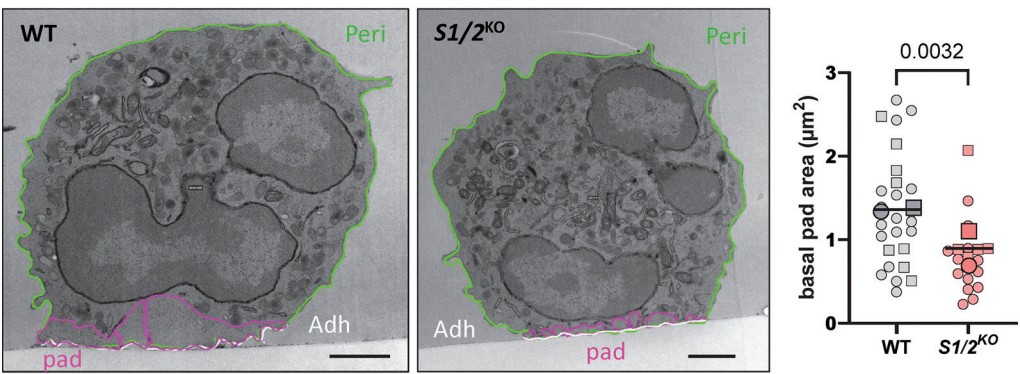

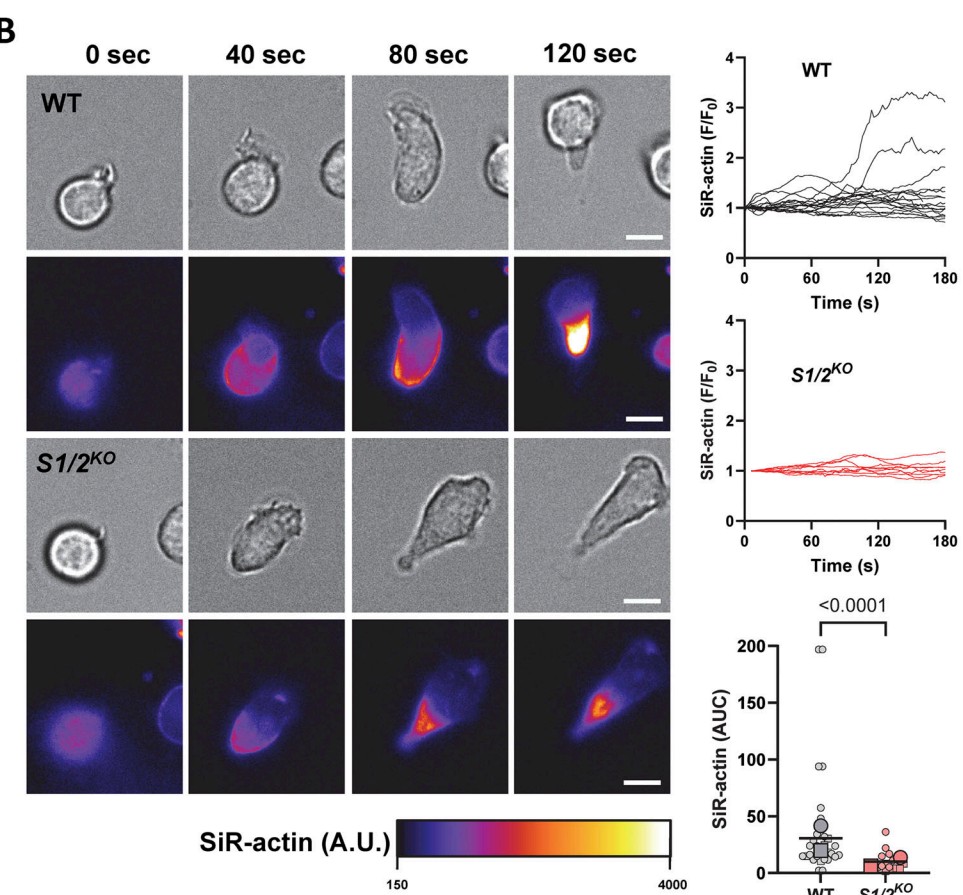

Figure 7.    **Effect of Stim1/2 ablation on actin dynamics at neutrophils adhesion sites. (A)** Representative electron micrographs of purified WT and *Stim1/2*KO adherent neutrophils (left) and quantification of the organelle-free area (presumably rich in actin) covering their adhesive membranes (right, *n* = 26/19 cells from two mice pairs, exact two-tailed Mann–Whitney U test). Adhesive (Adh) and nonadhesive membranes are outlined in white and green, respectively, cytosolic pad area in violet. Scale bar: 1 μm. Micrograph of *Stim1/2*KO neutrophil is duplicated from Fig. 5 A. **(B)** Time-lapse micrographs of flushed WT and *Stim1/2*-deficient neutrophils stained with the cell-permeable F-actin probe SiR-actin (left, Video 10 and Video 11) and change in SiR-actin fluorescence intensity during spreading (right, *n* = 31/18 cells from 2 WT/*Stim1/2*KO mice pairs, exact two-tailed Mann–Whitney U test). Scale bars: 5 μm. AUC, area under the curve.

**Online supplemental material**

Fig. S1 shows the gating strategy used to isolate neutrophils from BM, Salsa6f calibration, Ca²⁺ waves reported by Salsa6f, and the functional validation of Stim1/2 ablation, related to Fig. 1 and Fig. 2. Fig. S2 shows the effects of pharmacological inhibitors on

neutrophil Ca²⁺ responses, related to Fig. 3. Fig. S3 shows the effect of Stim1/2 ablation and PLC inhibition on neutrophil spreading, related to Fig. 4. Fig. S4 shows the effect of Stim1/2 ablation on ultrastructural parameters of mouse neutrophils, related to Fig. 5. Fig. S5 shows the effect of Stim1/2 ablation on

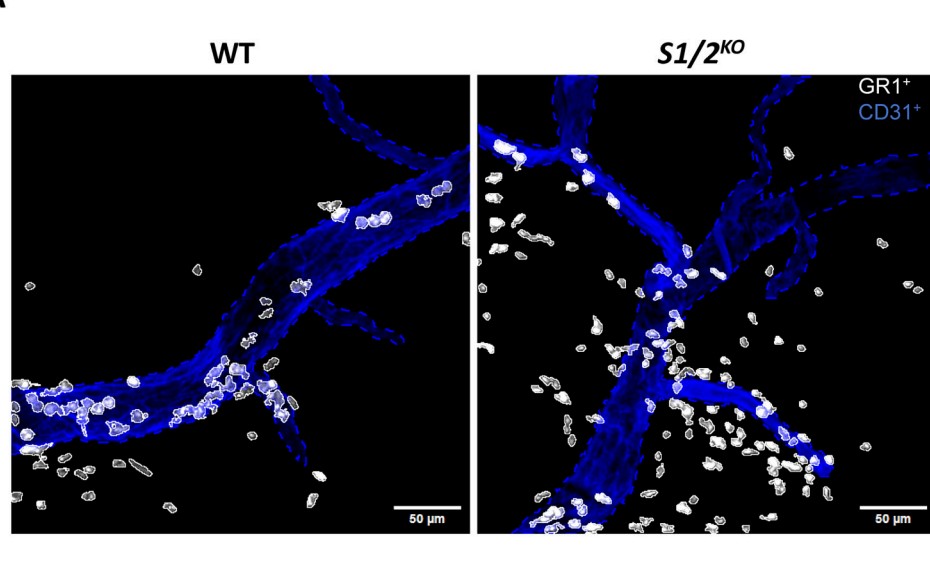

**A**

WT    S1/2^KO

GR1⁺
CD31⁺

50 μm    50 μm

**B**

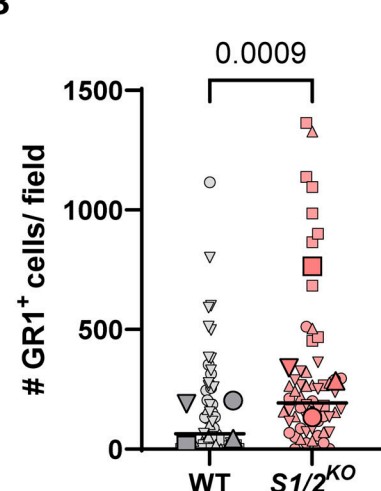

0.0009

Figure 8. **Effect of Stim1/2 ablation on neutrophil extravasation during inflammation. (A)** Fluorescence micrograph of cremaster tissue from WT and *Stim1/2^KO* mice, injected i.p. with 2.5 µg/ml TNF for 2 h prior to sacrifice and surgical extraction of the cremaster muscles. Granulocytes are stained white with GR-1, blood vessels blue with CD31 antibodies. **(B)** Quantification of interstitial GR-1⁺ cells within a distance of 75 µm from postcapillary venules in tissue sections from WT and *Stim1/2^KO* mice. *n* = 57/60 vessels from four mice pairs, exact two-tailed Mann–Whitney U test. Small and large symbols show data from individual vessels and mouse average, lines show median values by vessel.

the distribution of PMCA and IP$_3$R and the effect of Ca$^{2+}$ supplementation on the basal Ca$^{2+}$ levels of mouse neutrophils, related to Fig. 6. Fig. S6 shows the effect of Stim1/2 ablation on IP$_3$R mRNA levels and on neutrophil actin dynamics, related to Fig. 6 and Fig. 7. Table S1 shows the effects of soluble molecules and substrates on Ca$^{2+}$ signals during neutrophils spreading. Video 1 shows the time-lapse DIC and Salsa6f recordings in adherent mouse neutrophils, related to Fig. 1 C. Video 2 shows the time-lapse TIRF recordings of GCaMP6f fluorescence in an adherent mouse neutrophil, related to Fig. S1 D. Video 3 and Video 4 show the Salsa6f-based Ca$^{2+}$ recordings of adherent WT and *Stim1/2*-deficient neutrophils, related to Fig. 2 C. Video 5 shows the effect of PLC inhibition on Ca$^{2+}$ transients of adherent mouse neutrophils, related to Fig. 3 B. Video 6 and Video 7 show the time-lapse DIC recordings of WT and *Stim1/2*-deficient neutrophils plated to PLL, related to Fig. 4 B. Video 8 and Video 9 show the

TIRF recordings of CellMask-labeled WT and *Stim1/2*-deficient neutrophils plated to PLL, related to Fig. 4 C. Video 10 and Video 11 show the time-lapse DIC and SiR-actin recordings of WT and *Stim1/2*-deficient neutrophils plated to PLL, related to Fig. 7 B.

**Data availability**

The data that support the findings of this study are presented in the main and supplementary figures, with number of experiments and statistical tests applied. Primary data are available from the corresponding author upon reasonable request.

**Acknowledgments**

We thank Cyril Castelbou for the technical assistance, Elisa Husler for the help in image analysis, and the bioimaging, electron microscopy, flow cytometry, READS unit, and animal

core facilities of the Faculty of Medicine of the University of Geneva.

This work was funded by the Swiss National Foundation (grant number 310030_189042 [to N. Demaurex] and CRSK-3_221284 [to A. Carreras-Sureda]). Open Access funding provided by Université de Genève.

Author contributions: C. Rabesahala de Meritens: conceptualization, data curation, formal analysis, investigation, methodology, software, visualization, and writing—original draft, review, and editing. A. Carreras-Sureda: conceptualization, data curation, formal analysis, funding acquisition, methodology, supervision, and writing—review and editing. N. Rosa: formal analysis, investigation, methodology, and writing—review and editing. R. Pick: data curation, formal analysis, investigation, project administration, validation, and visualization. C. Scheiermann: formal analysis, investigation, and writing—review and editing. N. Demaurex: conceptualization, formal analysis, funding acquisition, investigation, methodology, project administration, supervision, validation, visualization, and writing—original draft, review, and editing.

Disclosures: The authors declare no competing interests exist.

Submitted: 12 June 2024

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

**Supplemental material**

Figure S1. **Validation of Salsa6f and of myeloid Stim1/2 ablation. (A)** Flow cytometry gating strategy to isolate tdT$^+$Ly6G$^+$CD115$^-$F4/80$^-$ neutrophils from flushed mouse BM. The two FACS profiles at bottom left are replicated in Fig. 1 A. Cartoon shows the LSL-Salsa6f cassette with its Ca$^{2+}$-insensitive tdTomato (tdT$^+$) and Ca$^{2+}$-sensitive GCaMP6f inserted at the Rosa26 locus. **(B)** Percentage of BM neutrophils isolated by negative selection from mice expressing or not Salsa6f. $n$ = 6 pairs of mice, two-tailed Welch's $t$ test. **(C)** Simultaneous Fura-2 and Salsa6f recordings in cells exposed to increasing extracellular [Ca$^{2+}$] in the presence of ionomycin (left) and steady-state Fura-2 and Salsa6f ratios as a function of intracellular [Ca$^{2+}$], calculated from [Ca$^{2+}$]$_{ext}$ using the Kd of Fura2 (right). **(D)** TIRF kymograph of Ca$^{2+}$ waves propagating in an adherent neutrophil (top) and changes in GCaMP6f intensity in three aligned regions (bottom, regions are indicated by dotted circles on the TIRF image) separated by 3.6 μm along the kymograph axis (dotted line). See Video 2. Representative of 10 recordings. **(E)** Flow cytometry Fluo-8 recordings and quantification of Ca$^{2+}$ responses evoked by Ca$^{2+}$ readmission to WT and $Stim1/2^{-/-}$ neutrophils exposed to Tg (1 μM) in Ca$^{2+}$-free medium. $n$ = 3 recordings from one pair of mice. **(F)** Salsa6f recordings and quantification of the Tg-Ca$^{2+}$ evoked responses in WT and $Stim1/2^{-/-}$ neutrophils. $n$ = 2 recordings from two pairs of mice. AUC, area under the curve. **(G)** Luminometric LO-12 recordings of the ROS production evoked by PMA in WT and $Stim1/2^{-/-}$ neutrophils. $n$ = 5/6 mice, in triplicate recordings.

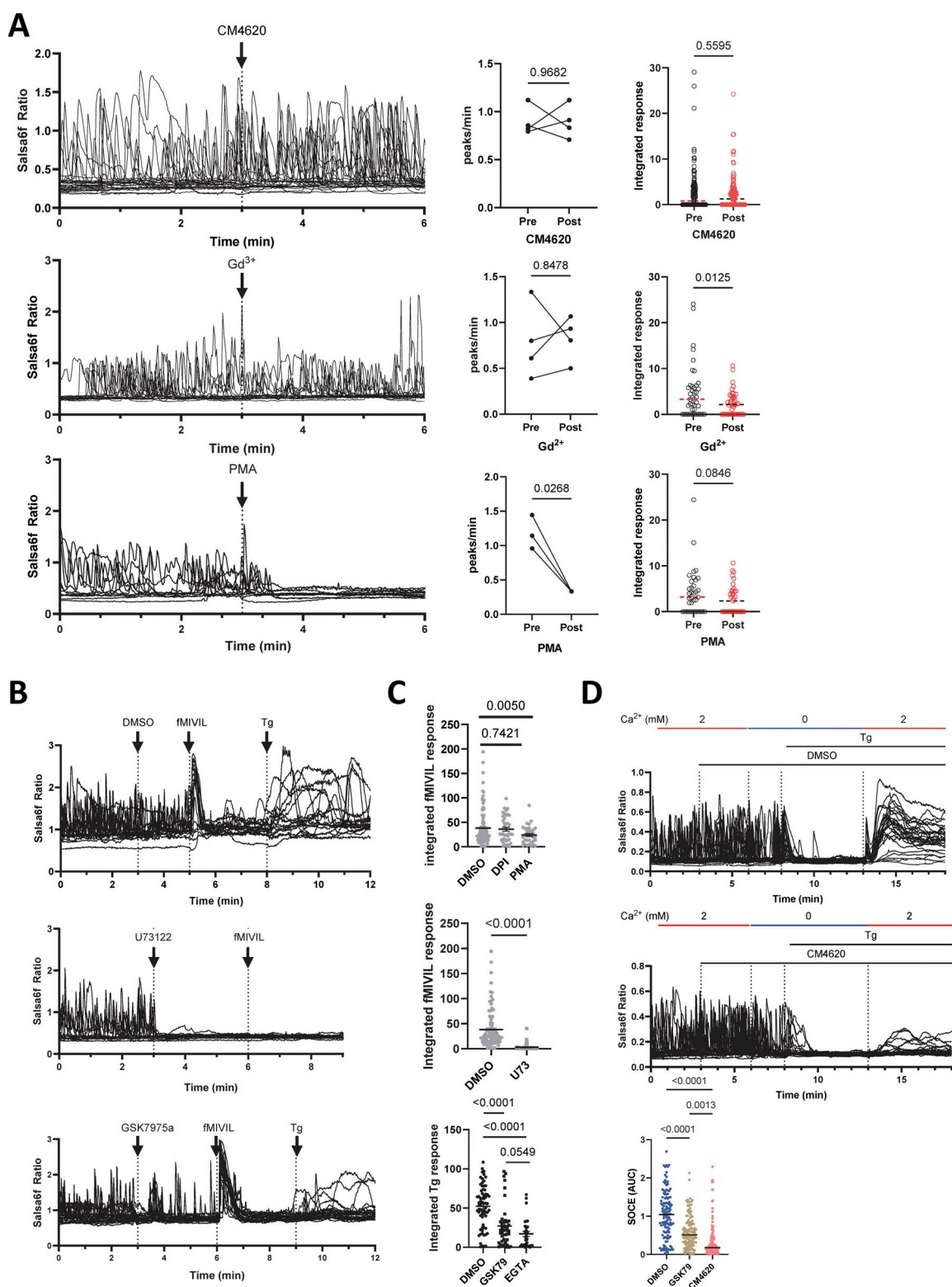

Figure S2. **Effect of pharmacological inhibitors on neutrophil Ca²⁺ responses. (A)** Salsa6f recordings (left), Ca²⁺ transient frequency (middle), and integrated Ca²⁺ responses (right) of adherent neutrophils exposed to CM4620 (10 µM), Gd³⁺ (10 µM), or to the PKC activator PMA (1 µM). n = 27/42/35 cells in 4/4/3 recordings from two mice, Student's paired two-tailed t test. **(B)** Entire recordings of Fig. 3 A showing the subsequent sequential addition of fMIVIL (10 nM) and Tg (1 µM) to cells treated with DMSO, GSK-7975a, and U73122. **(C)** Effect of the indicated inhibitors on Ca²⁺ responses evoked by fMIVIL and Tg. n = 92/40/35/45 cells for fMIVIL with DMSO/DPI/PMA/U73 and 68/44/31 cells for Tg with DMSO/GSK/EGTA from two to eight mice. Two-tailed Welch's t test. **(D)** Effect of DMSO, GSK7975a, and CM4620 on the Ca²⁺ responses evoked by Ca²⁺ readmission to cells treated with Tg in Ca²⁺-free medium. n = 125/126/116 cells in 4/4/4 recordings from two mice. Ordinary one-way ANOVA with Šídák multiple comparison test. Gd³⁺: gadolinium; fMIVIL: N-formyl-Met-Ile-Val-Ile-Leu; AUC: area under the curve.

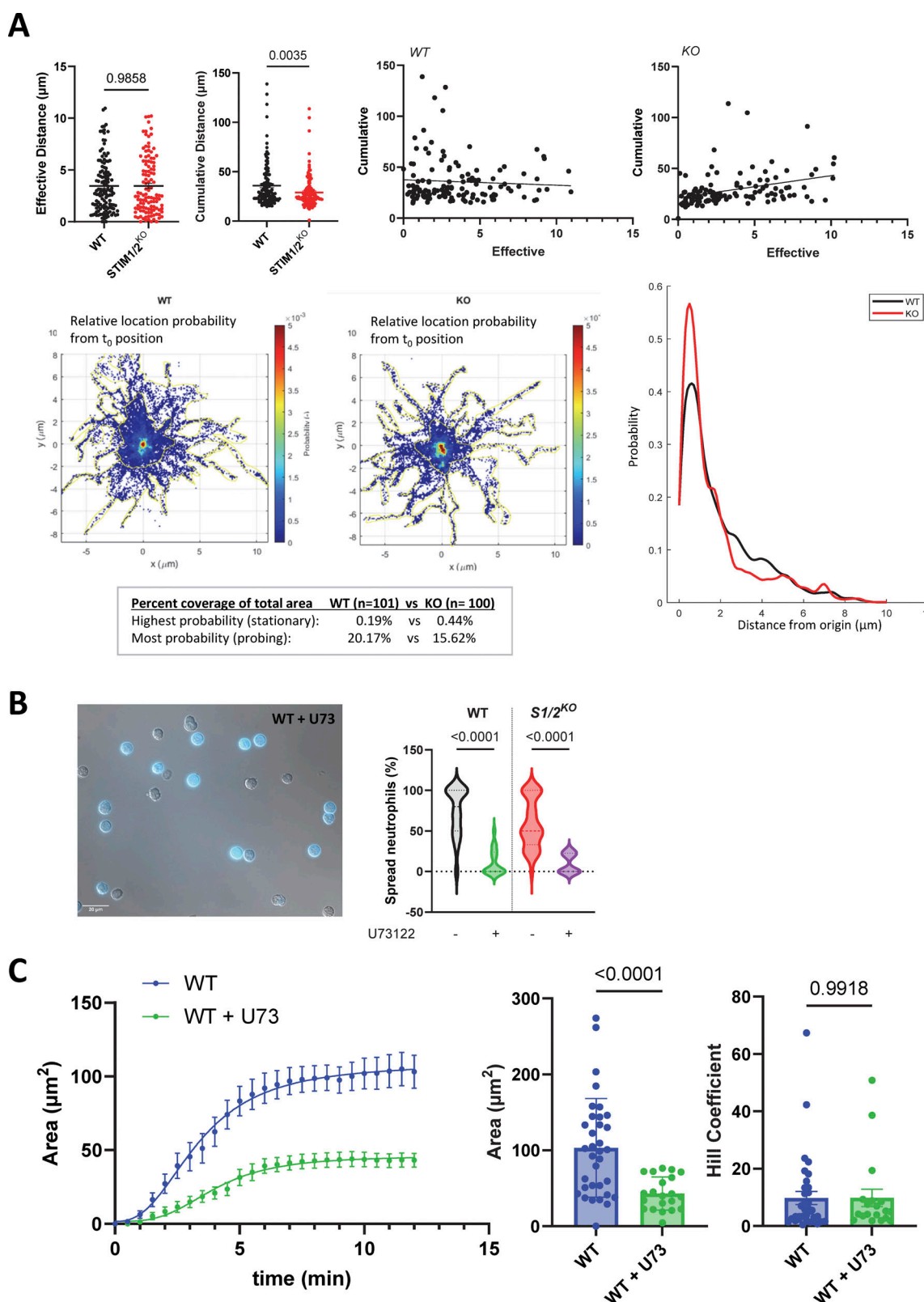

Figure S3. **Effect of Stim1/2 ablation on the spreading of neutrophils. (A)** Effective and cumulative distance covered by the centroids of WT and *Stim1/2⁻/⁻* neutrophils tracked in Fig. 4 A. Panels at right show the cross-correlation of these parameters. Bottom graphs show the centroid location and position relative to the initial landing spot and centroids probability distributions. **(B)** DIC micrographs (left) and fraction of spread WT and *Stim1/2⁻/⁻* neutrophils treated or not with U72133 following ~20-min plating at 37°C (right, *n* = 37/20 and 39/9 recordings from three WT/*Stim1/2⁻/⁻* mice pairs, two-tailed Welch's *t* test). **(C)** Change in the size of TIRF footprints of WT neutrophils treated or not with U72133 (*n* = 25/25 graphed and 34/19 analyzed cells from three WT/*Stim1/2⁻/⁻* mice pairs, two-tailed Welch's *t* test).

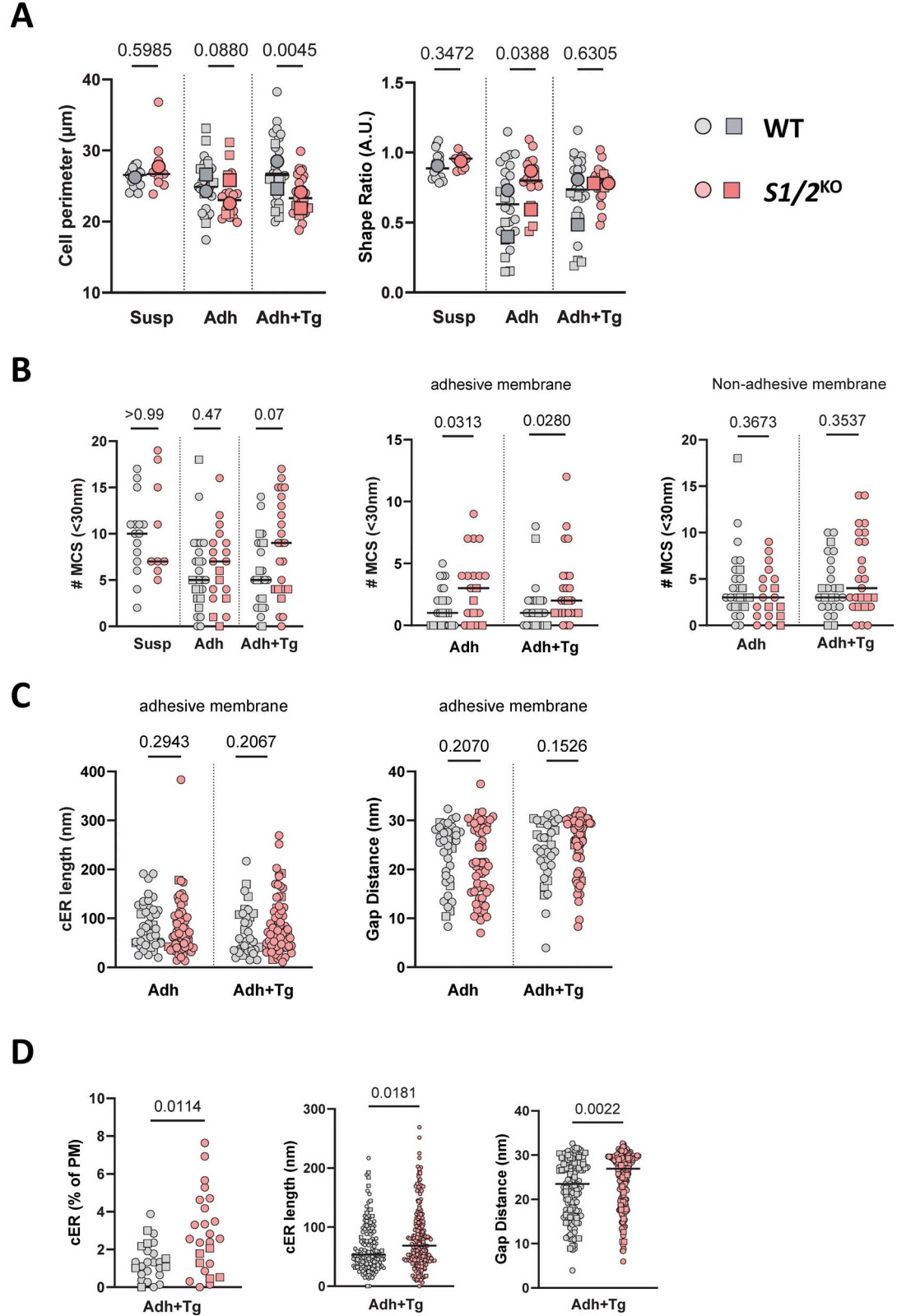

Figure S4. **Effect of Stim1/2 ablation on the ultrastructure of mouse neutrophils. (A)** Perimeter and shape factor of WT and *Stim1/2*⁻/⁻ neutrophils fixed in suspension or after adhesion to PLL in the absence or presence of Tg (Adh+Tg). **(B)** Number of contact sites detected in the indicated conditions along the entire cell perimeter (left) and on the adhesive and nonadhesive membranes (middle and right). **(C)** cER length and gap distance at adhesive PM of WT and *Stim1/2*⁻/⁻ neutrophils. *n* = 26/19 cells with 40/60 cER sheets from two mice pairs, Mann–Whitney U test. **(D)** cER proportion, length, and gap distance in adherent WT and *Stim1/2*⁻/⁻ neutrophils treated with Tg. *n* = 25/25 cells with 139/209 cER sheets pfrom two mice pairs, Mann–Whitney U test. Adh: adherent; Susp, suspended.

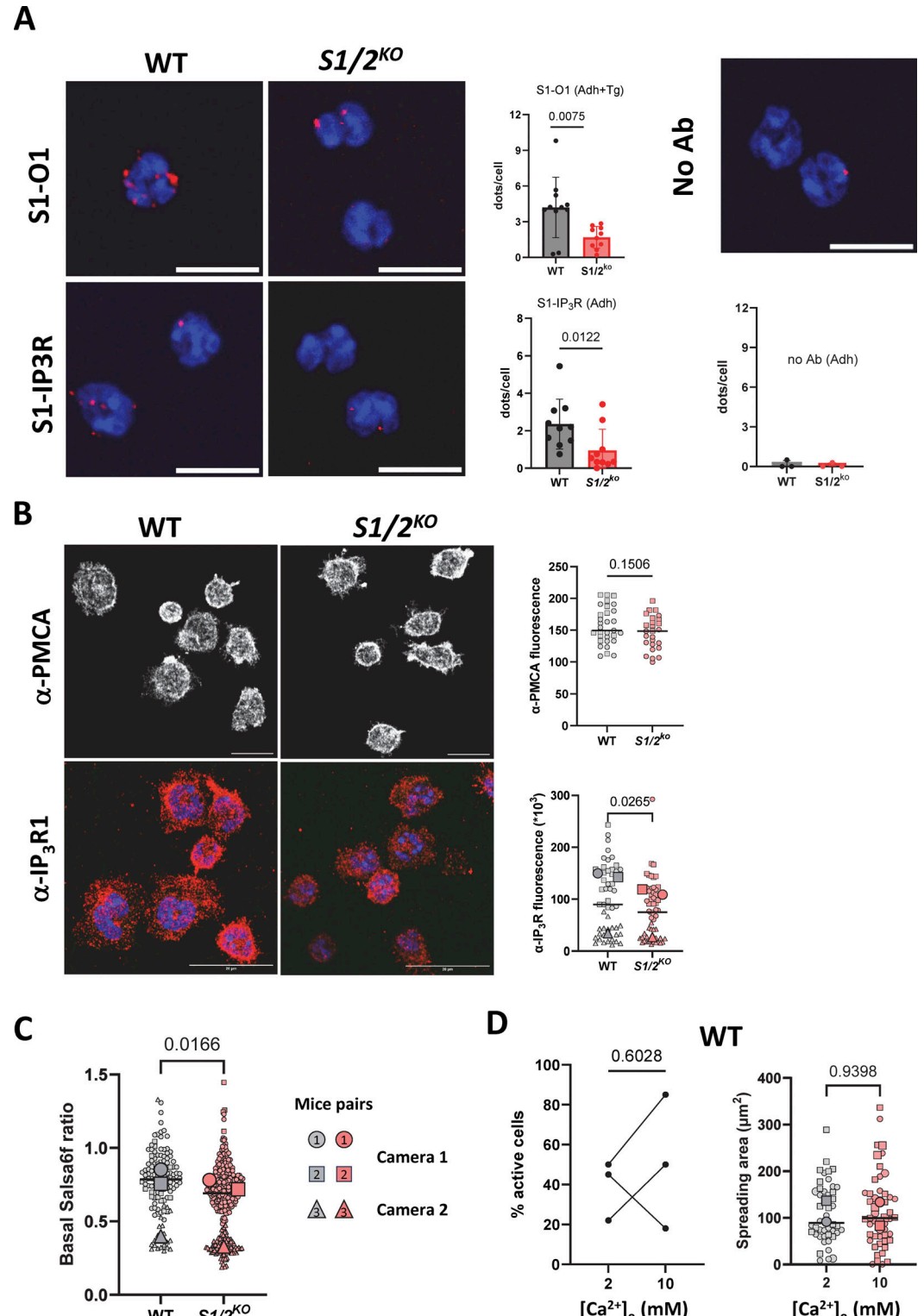

**Figure S5. Effect of Stim1/2 ablation on Ca²⁺ handling proteins and actin dynamics. (A)** Representative micrographs of PLAs between the indicated molecular targets in WT and *Stim1/2*-deficient neutrophils and in the absence of primary antibodies (left, scale bars: 10 µm) and quantification of the interactions (right, n = 11/10, 10/10 and 3/3 micrographs with >10 cells for each condition from 2/2/1 mice pairs, Mann–Whitney U test). **(B)** Confocal micrographs of WT and *Stim1/2⁻/⁻* neutrophils plated on PLL for 30 min, fixed and stained with an anti-PMCA antibody (left, nuclei stained blue with DAPI) and averaged cell-associated fluorescence area (right, n = 32/27 cells from one mice pair, Student's unpaired t test). **(C)** Basal Salsa6f ratio of WT and *Stim1/2*-deficient neutrophils recorded with two different imaging cameras. n = 108/236 and 37/87 cells from three mice pairs. Lines show median values, larger symbols mouse average. Data from camera 1 were converted to [Ca²⁺]cyt in Fig. 6 B. **(D)** Proportion of WT neutrophils exhibiting Ca²⁺ transients before and after switching from 2 to 10 mM [Ca²⁺]ext (left) and quantification of the spreading area of WT neutrophils adhered in 2 and 10 mM [Ca²⁺]ext (right, n = 22/33 cells in 4/5 recordings from two WT/*Stim1/2⁻/⁻* mice pairs, exact two-tailed Mann–Whitney U test on cell values).

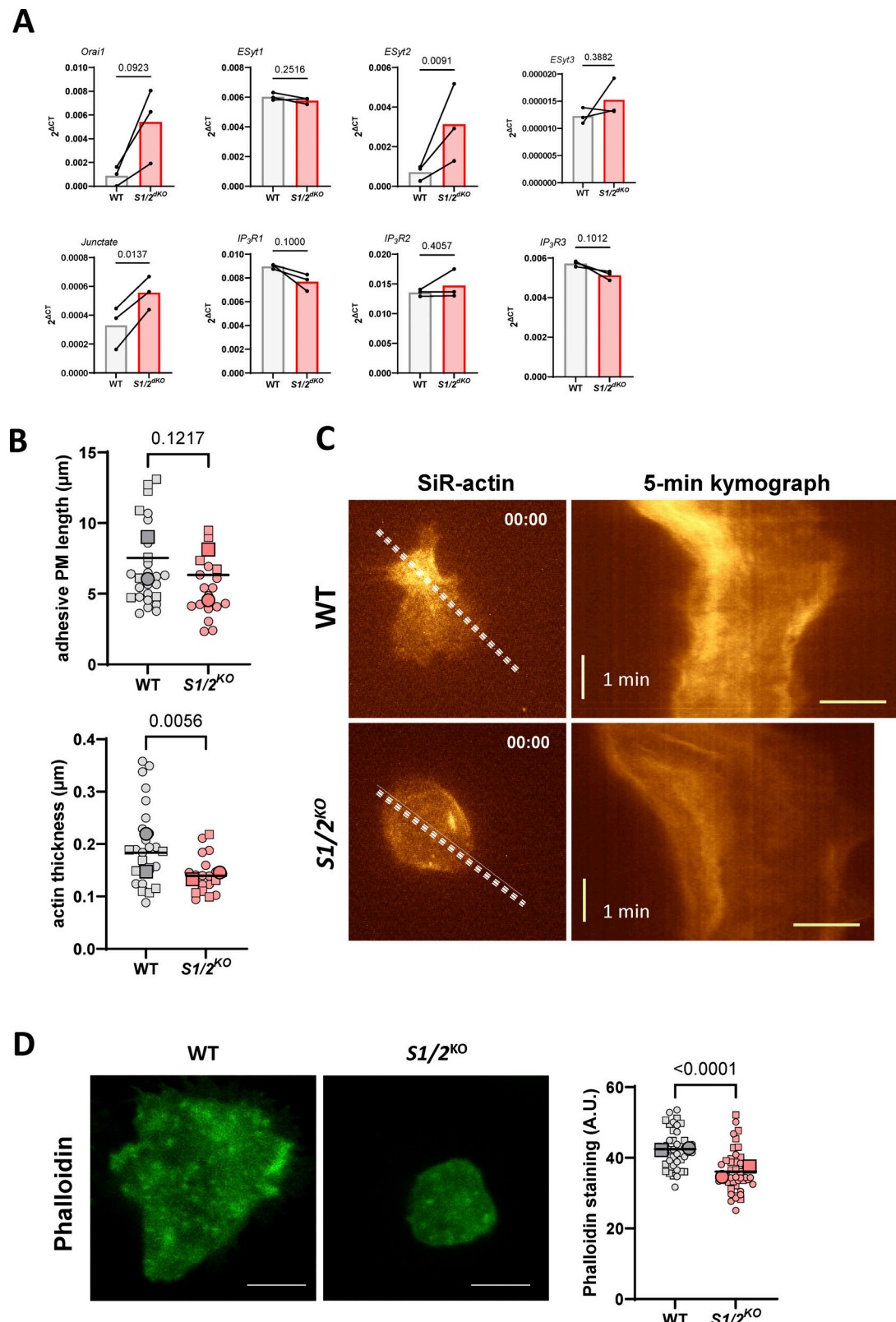

Figure S6. **The effect of Stim1/2 ablation on IP₃R mRNA levels and on neutrophil actin dynamics, related to** Fig. 6. **(A)** Real-time qPCR of neutrophils purified from WT and *Stim1/2*-deficient with primers designed against the indicated targets. $n = 3$ mice pairs, paired $t$ test. **(B)** Quantification of adhesive membranes length (left) and actin coating thickness (right) in the electron micrographs of Fig. 7 A. $n = 26/19$ cells from two mice pairs, exact two-tailed Mann–Whitney U test. **(C)** Sir-actin kymographs of WT and *Stim1/2*-deficient neutrophils during spreading. Dotted lines on micrographs at left indicate the kymograph axis. Scale bar: 10 μm. **(D)** Representative micrographs of spread WT and *Stim1/2*-deficient neutrophils stained with phalloidin (left) and quantification of phalloidin staining intensity (right, $n = 44/45$ cells in four experiments from two mice pairs, exact two-tailed Mann–Whitney U test).

Video 1.  **Ca²⁺ elevations in adherent mouse neutrophils.** Time-lapse differential interference contrast (DIC) and epifluorescence microscopy recording of neutrophils purified from the bone marrow of a *Cebpa^{cre/+};Salsa6f^{LSL}* mouse. Cells were plated on PLL and observed at 63x magnification. The green fluorescence of GCamP6f increases in intensity during Ca²⁺ elevations while the red fluorescence of TdTomato remains constant. Frames collected at 1 s intervals displayed at 20 frames/second. Related to Fig. 1 C.

Video 2.  **Ca²⁺ waves in adherent mouse neutrophils.** Time-lapse TIRF recording of a neutrophil purified from bone marrow of a *Cebpa^{cre/+};Salsa6f^{LSL}* mouse. Cells were plated on PLL and observed at 100× magnification. The green fluorescence of GCamP6f increases in intensity during Ca²⁺ elevations. Frames collected at 300 ms intervals displayed at 30.5 frames/second. Related to Fig. S1 D.

Video 3.  **Ca²⁺ elevations in adherent mouse neutrophils.** Time-lapse epifluorescence microscopy Salsa6f recording of bone marrow neutrophils from a *Cebpa^{cre/+};Salsa6f^{LSL}* mouse. Cells were plated on PLL and observed at 60× magnification. The green fluorescence of GCamP6f increases in intensity during Ca²⁺ elevations while the red fluorescence of TdTomato remains constant. Frames collected at 1 s intervals displayed at 40 frames/second. Related to Fig. 2 C.

Video 4.  **Effect of *Stim1/2* ablation on Ca²⁺ transients of adherent mouse neutrophils.** Time-lapse epifluorescence microscopy Salsa6f recording of bone marrow neutrophils from a *Cebpa^{cre/+};Stim1^{fl/fl};Stim2^{fl/fl};Salsa6f^{LSL}* mouse. Conditions as in Video 3. Related to Fig. 2 C.

Video 5.  **Effect of PLC inhibition on Ca²⁺ transients of adherent mouse neutrophils.** Time-lapse epifluorescence microscopy Salsa6f recording of bone marrow neutrophils from a *Cebpa^{cre/+};Salsa6f^{LSL}* mouse exposed to U73122 (1 μM) at t = 3 min. Conditions as in Video 3 and Video 4. Related to Fig. 3 B.

Video 6.  **Spreading of mouse neutrophils.** Time-lapse DIC recordings of bone marrow neutrophils from a *Cebpa^{+/+};Stim1^{fl/fl};Stim2^{fl/fl}* (WT) mouse. Cells were plated on PLL and observed at 63× magnification. Frames collected at 2 s intervals displayed at 30 frames/second. Related to Fig. 4 B.

Video 7.  **Effect of *Stim1/2* ablation on the spreading of mouse neutrophils.** Time-lapse DIC recordings of bone marrow neutrophils from a *Cebpa^{cre/+};Stim1^{fl/fl};Stim2^{fl/fl}* mouse. Conditions as in Video 6. Related to Fig. 4 B.

Video 8.  **Lamellipodia formation during mouse neutrophils spreading.** Time-lapse TIRF recording of CellMask-labelled neutrophils purified from bone marrow of a *Cebpa^{+/+};Stim1^{fl/fl};Stim2^{fl/fl}* (WT) mouse. Cells were plated on PLL and observed at 100× magnification. Frames collected at 30 s intervals displayed at 4 frames/second. Related to Fig. 4 C.

Video 9.  **Effect of *Stim1/2* ablation on lamellipodia formation during mouse neutrophils spreading.** Time-lapse TIRF recording of CellMask-labelled neutrophils purified from bone marrow of a *Cebpa^{cre/+};Stim1^{fl/fl};Stim2^{fl/fl}* mouse. Conditions as in Video 8. Related to Fig. 4 C.

Video 10.  **Actin dynamics during mouse neutrophils spreading.** Time-lapse DIC and SiR-actin recording of bone marrow neutrophils from a *Cebpa^{+/+};Stim1^{fl/fl};Stim2^{fl/fl}* (WT) mouse. Cells were labelled with Sir-Actin, plated on PLL, and observed at 63× magnification. The red fluorescence of SiR-actin increases in intensity as actin monomers assemble into microfilaments. Frames collected at 2 s intervals displayed at 8 frames/second. Related to Fig. 7 B.

Video 11.  **Effect of *Stim1/2* ablation on actin dynamics during mouse neutrophils spreading.** Time-lapse DIC and SiR-actin recording of bone marrow neutrophils from a *Cebpa^{cre/+};Stim1^{fl/fl};Stim2^{fl/fl}* mouse. Conditions as in Video 10. Related to Fig. 7 B.

**Provided online is Table S1. Table S1 shows the effects of soluble molecules and substrates on Ca²⁺ signals during neutrophils spreading.**

