## [Peer Review File · The Journal of Cell Biology]

STIM1/2 maintain signaling competence at ER-PM contact sites during neutrophil spreading

Camille Rabesahala de Meritens, Amado Carreras Sureda, Nicolas Rosa, Robert Pick, Christoph Scheiermann, and Nicolas Demaurex

Corresponding Author(s): Nicolas Demaurex, University of Geneva

Review Timeline:

Submission Date:	2024-06-12
Editorial Decision:	2024-08-06
Revision Received:	2024-11-26
Editorial Decision:	2025-01-14
Revision Received:	2025-02-05

Monitoring Editor: Tamas Balla

Scientific Editor: Tim Fessenden

Transaction Report:

DOI: <https://doi.org/10.1083/jcb.202406053>

August 6, 2024

Re: JCB manuscript #202406053

Prof. Nicolas Demaurex
University of Geneva
Cell Physiology
Ch de Charvel 11
Vésenaz 1222
Switzerland

Dear Prof. Demaurex,

Thank you for submitting your manuscript entitled "STIM1/2 maintain signaling competence at ER-PM contact sites during neutrophil spreading". Your manuscript has been assessed by expert reviewers, whose comments are appended below. We sincerely regret the long duration for the reviewing process and subsequent editorial decision, due to both reviewer and editor availability, and we appreciate your patience. Although the reviewers express potential interest in this work, significant concerns unfortunately preclude publication of the current version of the manuscript in JCB.

While reviewers were mixed, two commended the intriguing observations on the involvement of STIM1/2 and ER-plasma membrane contacts in regulating neutrophil spreading. All reviewers felt that important details into the mechanism by which STIM1/2 act in this setting were missing, in particular concerning the connection with actin-rich adhesive membranes and actin remodeling. Related to this, Reviewers 2 and 3 felt the evidence for interactions among IP3R, STIM, and Orai should be strengthened, including by methods other than by proximity ligation where possible. We concur with Reviewer 1 that these findings would be considerably strengthened by *in vivo* studies to confirm the role of STIM1/2 in neutrophil behaviors during infection. Finally, Reviewer 3 requested a rescue experiment to confirm the importance of STIM1/2 in neutrophil spreading. While all reviewer points must be addressed in some form, additional data beyond those noted here are not required in a revision.

Please let us know if you are able to address the major issues outlined above and wish to submit a revised manuscript to JCB. Note that a substantial amount of additional experimental data likely would be needed to satisfactorily address the concerns of the reviewers. The typical timeframe for revisions is three to four months. While most universities and institutes have reopened labs and allowed researchers to begin working at nearly pre-pandemic levels, we at JCB realize that the lingering effects of the COVID-19 pandemic may still be impacting some aspects of your work, including the acquisition of equipment and reagents. Therefore, if you anticipate any difficulties in meeting this aforementioned revision time limit, please contact us and we can work with you to find an appropriate time frame for resubmission. Please note that papers are generally considered through only one revision cycle, so any revised manuscript will likely be either accepted or rejected.

If you choose to revise and resubmit your manuscript, please also attend to the following editorial points. Please direct any editorial questions to the journal office.

GENERAL GUIDELINES:

Text limits: Character count is < 40,000, not including spaces. Count includes title page, abstract, introduction, results, discussion, and acknowledgments. Count does not include materials and methods, figure legends, references, tables, or supplemental legends.

Figures: Your manuscript may have up to 10 main text figures. To avoid delays in production, figures must be prepared according to the policies outlined in our Instructions to Authors, under Data Presentation, <https://jcb.rupress.org/site/misc/ifora.xhtml>. All figures in accepted manuscripts will be screened prior to publication.

IMPORTANT: It is JCB policy that if requested, original data images must be made available. Failure to provide original images upon request will result in unavoidable delays in publication. Please ensure that you have access to all original microscopy and blot data images before submitting your revision.

Supplemental information: There are strict limits on the allowable amount of supplemental data. Your manuscript may have up to 5 supplemental figures. Up to 10 supplemental videos or flash animations are allowed. A summary of all supplemental material should appear at the end of the Materials and methods section.

Please note that JCB now requires authors to submit Source Data used to generate figures containing gels and Western blots with all revised manuscripts. This Source Data consists of fully uncropped and unprocessed images for each gel/blot displayed in the main and supplemental figures. Since your paper includes cropped gel and/or blot images, please be sure to provide one

Source Data file for each figure that contains gels and/or blots along with your revised manuscript files. File names for Source Data figures should be alphanumeric without any spaces or special characters (i.e., SourceDataF#, where F# refers to the associated main figure number or SourceDataFS# for those associated with Supplementary figures). The lanes of the gels/blots should be labeled as they are in the associated figure, the place where cropping was applied should be marked (with a box), and molecular weight/size standards should be labeled wherever possible.

If you choose to resubmit, please include a cover letter addressing the reviewers' comments point by point. Please also highlight all changes in the text of the manuscript.

Regardless of how you choose to proceed, we hope that the comments below will prove constructive as your work progresses. We would be happy to discuss them further once you've had a chance to consider the points raised. You can contact the journal office with any questions at cellbio@rockefeller.edu.

Thank you for thinking of JCB as an appropriate place to publish your work.

Sincerely,

Tamas Balla
Monitoring Editor
Journal of Cell Biology

Tim Fessenden
Scientific Editor
Journal of Cell Biology

Reviewer #1 (Comments to the Authors (Required)):

De Meritens et al. use genetic ablation of *stim1/2* in the myeloid lineage to study the effect of their loss on neutrophil migration. In a set of carefully control studies they show effects on Ca^{2+} signaling in neutrophils as well as on their migration following the loss of *stim1/2*. They further interestingly show that the migration phenotype correlates with the broad distribution of ER-PM membrane contact sites (MCS) at the base of the cell whereas in WT neutrophils the MCS localize specifically to the rear end of the migrating neutrophil whereas the front end is actin rich and devoid of MCS. Although these studies are quite interesting there is no mechanistic validation of the findings. Surprisingly the loss of STIM1/2 and the relationship of the actin dense leading edge is not investigated in any details. Are STIM1/2 required for the formation of the actin dense leading edge? Stim2 has been shown to regulate basal Ca^{2+} levels and authors find a decrease in resting Ca^{2+} in neutrophils although they seem to indicate that it's due to re-localization of IP3R? Does the lower resting Ca^{2+} affect actin fiber formation? Is Ca^{2+} evenly distributed across migrating cells? This last question is important as a Ca^{2+} gradient has been shown in migrating cells. The authors touch on these possibilities in the discussion but they don't test these ideas. This includes potential increased expression/function of ER-PM tethers in response to STIM1/2 KO. This is relatively simple at least in terms of checking expression levels of the different tethers. Proteomic studies on the STIM1/2 KO neutrophils could be quite informative. Appropriately the authors highlight the reversible tethering role of STIM1 in terms of lipid binding and Ca^{2+} dependency, but in the absence of STIM1 one would expect weaker tethering which would correlate with less MCS which is not observed. So it is not clear to me how loss of STIM1/2 increases MCS at the adhesion leading edge if not through modulating the actin cytoskeleton. Finally, and most importantly, it would be vital to show a physiological neutrophil defect in terms of fighting infections in the myeloid STIM1/2 KO mice to correlate the migration defects and MCS phenotypes to a physiological response. Another interesting question that is not addressed is whether the observed phenotype is STIM1 or STIM2 dependent or requires the loss of both proteins. In summary, these are intriguing studies that could be quite impactful but currently lack the physiological context as well as the mechanistic basis of the observed phenotype.

Specific Comments:

The tables are informative, but it would be best to list actual values (means with SE and p values) rather than qualitative measure only.

For the inhibitors studies it would be informative to show their effects on a standard SOCE experiment in neutrophils (similar to S2A) to confirm and quantify that at the concentrations used there is significant inhibition of SOCE in WT neutrophils as this has

important impact on the conclusion that STIM1/2 affect migration independently of their ability to modulate Ca²⁺ influx. Please provide more details for Fig. 4B. Is the % of spread neutrophils analysis performed on a per dish basis? If that is the case then some dishes of STIM1/2 KO cells have 100% spread neutrophils similar to WT? that would be unexpected. Also, from the images shown it seems unlikely to get 100% spreading even in the WT.

Fig. 4B the number of cells analyzed for the STIM1/2 KO for cER is rather small (only 9 cells). As there is a trend in the current dataset toward less cER in the STIM1/2 Kos increasing the number of analyzed cells is advisable.

For the violin graphs it is difficult to visualize the mean, please indicate with a thicker bar or some other method to make it easier to visualize.

The conclusion in 241-2 "Unexpectedly, Stim1/2 ablation did not alter the proportion and morphology of contact sites but instead prevented their retrieval from adhesive membranes." May be too simplistic. It is not that the MCS are not retrieved from adhesive membrane as the lagging edge of the cell in WT still has MCS on the adhesive side of the membrane. It's rather that in the absence of stim1/2 the actin rich regions on the leading edge are no longer formed. Now the relationship between the MCS removal from the adhesive leading edge and the formation of actin rich regions is likely more complex.

The data in Fig. S5B are important for the conclusions and the future studies. Suggest moving them to the main manuscript.

The proximity ligation experiments are not sufficient on their own and would need independent validation somehow to support the conclusion of the redistribution of IP3 receptors away from the plasma membrane, such as co-IP or immunogold localization.

Reviewer #2 (Comments to the Authors (Required)):

This manuscript addresses the role of STIM1 and STIM2 proteins in murine neutrophil spreading, important for pathogen clearance. The authors found that STIM proteins are important for calcium signals generated during neutrophil spreading using a transgenic mouse model expressing the ratiometric calcium sensor Salsa6f. Consistent with a role of calcium signals in neutrophil spreading, they found reduced spreading of neutrophils without STIM1/2. Next, they found calcium signals generated during neutrophil spreading can be blocked by U73122, a PLC inhibitor, but not by GSK-7975a, an Orai1 inhibitor. Based on these results, they claimed that calcium entry is not required for calcium activity of adherent neutrophils. They further examined ER-PM contact sites in WT and STIM1/2 DKO neutrophils using electron microscopy and used PLA assays to compare proximity between ER and PM in WT vs STIM1/2 DKO neutrophils.

This manuscript has several major flaws and the results do not warrant the conclusions summarized in the abstract.

1. Figure 3. The role of Orai1 in generating calcium signals in neutrophils was not rigorously tested. The claim that calcium entry is not required for calcium activity of adherent neutrophils is based on the effect of only one Orai1 inhibitor.
2. Figure 5. The authors claimed that "ER-PM contact sites were dynamically excluded from actin-rich adhesive membranes in WT, but not Stim1/2-deficient neutrophils, linking the persistence of signaling-deficient contact sites to impaired actin remodeling". Nevertheless, the authors did not show or analyze "actin-rich adhesive membranes" in this manuscript. Also, the concept that ER-PM contact sites are excluded from actin-rich membrane is not new.
3. Figure 5. The authors emphasize that "unexpectedly (STIM1/2 ablation) did not impact the proportion, proximity, and length of ER-PM contact sites". However, many differences between WT and STIM1/2 DKO, especially at the adhesive membrane where cell spreading occurs, were shown in Figure S5B and S5C. There is a bias toward the conclusion the authors would like to make.
4. Figure 6A. It does not make sense to perform PLA experiments between S1-O1 or S1-IP3R in S1/S2 DKO cells. Also, more experiments are needed to rigorously test the hypothesis that "ER Ca²⁺ release channels are redistributed away from the PM" in STIM1/2 DKO cells in addition to counting dots per cell from IP3R-PMCA PLA experiments.
5. Figure 6B. It has been shown previously that STIM2 deficiency results in reduced basal calcium (PMID: 18160041)

Reviewer #3 (Comments to the Authors (Required)):

In the study "STIM1/2 maintain signaling competence at ER-PM contact sites during neutrophil spreading" by Rebesch de Meritens et al, the authors present evidence that myeloid-specific ablation of Stim1/2 prevents adhesion-induced elevations and impairs neutrophil spreading. The authors use quantitative electron microscopy to establish that ER-PM contacts were not dependent on STIM1/2 expression and use proximity ligation assays (PLA) to demonstrate that the ablation of STIM1/2 expression negatively impacts IP3R association with Ca-pumps (Ca²⁺ ATPase (PMCA). Although the studies are generally sound and use state-of-the-art technological approaches, some of the experiments seem descriptive in nature. Furthermore, expansion of the sample size in some experiments and complementation with genotype reversal approaches (rescue experiments) would help solidify some of the concepts and interpretations.

Major comments

1. myeloid-specific Stim1/2 ablation prevents adhesion-induced Ca²⁺ elevations and impairs the spreading of mouse neutrophils. at least one rescue experiment would help validate the role of STIM1 or 2 in these processes.
2. Using pharmacological inhibitors, the authors conclude that the inhibitory effect of Stim1/2 depletion on calcium/spreading is recapitulated by PLC inhibition, leading to the conclusion that "... Stim1/2 ablation impairs the PLC-dependent spreading of

neutrophils". However, in Figure S4B the authors show that the PLC inhibitor U73122 inhibits spreading not only in WT but also in Stim1/2-KO cells suggesting that PLC regulates spreading by interfering with mechanisms independent of STIM1/2. This should be discussed.

3. Proximity ligation assay is a powerful technique however, S1-O1 interaction is visualized in Stim1/2-KO, with many cells showing as many as 3 dots/cell. Does the author consider this to be the background level? For reference, a similar number of interactions are observed in WT cells for S1-IP3R. The authors should complement these studies with colocalization analyses of endogenous proteins in 3-D. Also, additional information should be added to the methods section, specifying that PLA was analyzed in 3-D as well as the step used in this analysis.

4. Define the abbreviations in figure legends, for example, S1 in Figure 6.

5. In some experiments (for example Fig. 6 but also in many others), only "ten cells from 2 pairs are analyzed". The authors should expand the "n, (number of mice)" in key experiments and data presented as super-plots whenever possible.

Minor comments:

1. The author should clearly indicate where they use post-hoc neutrophil identification or isolation of neutrophils by negative selection.

2. The authors wrote "Inspection of the electron micrographs indicated that WT neutrophils had a long adhesive membrane covered by a thick actin cytoskeleton that was largely devoid of contact sites (Fig. 5A)." This should be rephrased as the thick actin cytoskeleton is not visualized in these micrographs.

3. The putative reduced immunoreactivity of IP3R in S1/2Ko (Fig. S6B) is too mild and the use of n=1 mouse pair raises questions about whether this difference is real. Expanding the n and using super-plots would help establish whether there are biological differences in IP3R levels.

Response to Reviewers Comments

We thank the Editors for the professional handling of our manuscript and the Reviewers for their insightful and constructive suggestions. Below is a point-by-point response to the critiques raised. For each comment the input of the Reviewer is listed in *italics* followed by our response in plain letters. In the interest of conciseness, only critical comments that require a response are listed. The page numbers refer to the revised manuscript with changes highlighted in track mode

Reviewer #1

Although these studies are quite interesting there is no mechanistic validation of the findings. Surprisingly the loss of STIM1/2 and the relationship of the actin dense leading edge is not investigated in any details. Are STIM1/2 required for the formation of the actin dense leading edge?

Good point. We have quantified the formation of actin filaments during neutrophils spreading by light and electron microscopy. Of note, the actin-rich structures forming at the cell-substrate interface are better described as lamellipodia than leading edge since neutrophils are not polarized as they spread. *Stim1/2*-deficient neutrophils had reduced phalloidin staining at adhesion sites, impaired SiR-actin dynamics during spreading, and a reduced actin-dense coverage of adhesive membranes at the ultrastructural level (Fig. 7). These data mechanistically link *Stim1/2* ablation to impaired actin dynamic and we thank the reviewers for suggesting these experiments.

Stim2 has been shown to regulate basal Ca²⁺ levels and authors find a decrease in resting Ca²⁺ in neutrophils although they seem to indicate that it's due to re-localization of IP3R? Does the lower resting Ca²⁺ affect actin fiber formation?

We do not know whether the reduced resting Ca²⁺ levels are linked to alterations on IP₃R or to the loss of Stim2. We are currently generating mice bearing a targeted disruption in the Stim2 gene, but the generation of these animals is taking longer than anticipated and their description would considerably delay the resubmission of this manuscript. Earlier studies with Ca²⁺ chelators reported that actin fiber formation is reduced at low Ca²⁺ levels, and our new data (Fig. 7) indicate that a reduction in basal Ca²⁺ activity and resting levels is associated with impaired actin dynamics in *Stim1/2*-deficient neutrophils (see above). To test the causality between Ca²⁺ and actin fiber formation, we increased the external Ca²⁺ concentration to supraphysiological levels, from 2 to 10 mM. Remarkably, Ca²⁺ supplementation restored basal Ca²⁺ levels, adhesion-induced Ca²⁺ elevations, and spreading competence in *Stim1/2*-deficient neutrophils (Fig. 6). These data confirm that Ca²⁺ levels regulate neutrophils actin-based motility.

Is Ca²⁺ evenly distributed across migrating cells? This last question is important as a Ca²⁺ gradient has been shown in migrating cells.

Ca²⁺ waves were previously reported in migrating human neutrophils and associated with the remodeling of the actin cytoskeleton (Jaconi et al., 1991). In our case, a gradient is difficult to image since neutrophils are not polarized during spreading and a leading edge not always detectable. Using TIRF time-lapse imaging of Salsa6f, we could record Ca²⁺ waves propagating along the cell axis (Fig. S1C), but cannot relate them to directionality as the cells are not performing directed migration.

The authors touch on these possibilities in the discussion but they don't test these ideas. This includes potential increased expression/function of ER-PM tethers in response to STIM1/2 KO. This is relatively simple at least in

terms of checking expression levels of the different tethers. Proteomic studies on the STIM1/2 KO neutrophils could be quite informative.

We now document by PCR the expression levels of the different tethers. The loss of Stim1/2 was associated with an increased expression of junctate and of E-Syt2 (Fig. S6E).

So it is not clear to me how loss of STIM1/2 increases MCS at the adhesion leading edge if not through modulating the actin cytoskeleton.

We believe that the increased expression of other tethering proteins (junctate and E-Syt2) might account for the persistent contact sites forming on adhesive membranes of spreading *Stim1/2*-deficient neutrophils.

Finally, and most importantly, it would be vital to show a physiological neutrophil defect in terms of fighting infections in the myeloid STIM1/2 KO mice to correlate the migration defects and MCS phenotypes to a physiological response.

We have quantified the recruitment of neutrophils to sites of inflammation *in vivo*. Following the injection of TNF in the mouse cremaster muscle, more neutrophils were recruited to the site of inflammation 2h post-injection in mice lacking Stim1/2 in myeloid cells (Fig. 8). These unexpected data indicate that the Stim1/2-dependent Ca^{2+} signals that facilitate neutrophil spreading *in vitro* mitigate neutrophil migration *in vivo*. Neutrophil extravasation involves a sequence of rolling, adhesion and transmigration events and the amounts of neutrophils accumulating in inflamed tissue reflect the balance between forward and reverse transmigration. Additional *in vivo* experiments are required to establish how Stim1/2 deficiency impacts the different steps in the leukocyte adhesion cascade and whether the role of Stim1/2 may differ between static conditions or under flow. This is now discussed in the text (lines 394-6).

Another interesting question that is not addressed is whether the observed phenotype is STIM1 or STIM2 dependent or requires the loss of both proteins.

We agree that this is an important question. Since primary neutrophils cannot be genetically manipulated (their lifespan is only a couple of hours *in vitro*) we cannot re-express STIM1 or STIM2 in our double knock-out neutrophils. We are currently generating mice bearing a targeted disruption in either *Stim1* or *Stim2* but since this task cannot be performed within the timeframe of the revision we plan to report the phenotype of these single KO mice in a subsequent study.

Specific Comments:

The tables are informative, but it would be best to list actual values (means with SE and p values) rather than qualitative measure only. Done

For the inhibitors studies it would be informative to show their effects on a standard SOCE experiment in neutrophils (similar to S2A) to confirm and quantify that at the concentrations used there is significant inhibition of SOCE in WT neutrophils as this has important impact on the conclusion that STIM1/2 affect migration independently of their ability to modulate Ca^{2+} influx.

Fig. S3D now show the effect of the inhibitors on a standard Tg-readmission protocol. GSK-7975A and CM4620 reduced SOCE by 43% and 65%, respectively, and did not alter the frequency or integrated responses of the adhesion-induced Ca^{2+} fluctuations (Fig. 3A and S3A).

Please provide more details for Fig. 4B. Is the % of spread neutrophils analysis performed on a per dish basis? If that is the case then some dishes of STIM1/2 KO cells have 100% spread neutrophils similar to WT? that would be unexpected. Also, from the images shown it seems unlikely to get 100% spreading even in the WT.

The percentage of spread neutrophils was determined by taking a series of micrographs 3 min after plating bone marrow cells on PLL-coated coverslips. 3 pairs of WT/KO mice were compared head-to-head in 3 different days, using 2-3 coverslips for each mouse to get 11-20 micrographs per animal. Using the Ly6G staining to identify neutrophils, the percentage of spread cells was then determined on each micrograph. There were indeed fields with 100% spread neutrophils, in both genotypes. As requested by reviewer 3, we now express the data as superplots whenever possible. This transparent presentation mode highlights the day-to-day variability in neutrophil spreading efficiency (likely reflecting differences in neutrophil priming during the extraction procedure) and confirms the reproducibility of the spreading defect associated with Stim1/2 deficiency (new Fig. 4B).

Fig. 4B the number of cells analyzed for the STIM1/2 KO for cER is rather small (only 9 cells). As there is a trend in the current dataset toward less cER in the STIM1/2 Kos increasing the number of analyzed cells is advisable.

The referred Fig. is 6B, not 4B. The low sampling (n=9) was only for the suspended cells. We have now analyzed all the cells on the EM grids. The amount of cER was comparable between the two genotypes (new Fig. 6B).

For the violin graphs it is difficult to visualize the mean, please indicate with a thicker bar or some other method to make it easier to visualize.

Done. We now show in superplots the proportion of cortical ER by cell and in violin graphs the ultrastructural parameters of individual MCS (length and gap distance)

The conclusion line 241-2 "Unexpectedly, Stim1/2 ablation did not alter the proportion and morphology of contact sites but instead prevented their retrieval from adhesive membranes." May be too simplistic. It is not that the MCS are not retrieved from adhesive membrane as the lagging edge of the cell in WT still has MCS on the adhesive side of the membrane. It's rather that in the absence of stim1/2 the actin rich regions on the leading edge are no longer formed. Now the relationship between the MCS removal from the adhesive leading edge and the formation of actin rich regions is likely more complex.

We agree that the sentence was too simplistic as this is a dynamic situation with MCS, Ca²⁺, and actin all impacting each other. The persistence of MCS on adhesive membranes of Stim1/2-deficient cells thus also likely reflects the reduced formation of actin filaments on this membrane, due to the lack of Ca²⁺ signals, and not merely the inability of MCS lacking Stim1/2 to detach, as we implied. We have rephrased this sentence to "persisted on adhesive membranes" (line 257)

The data in Fig. S5B are important for the conclusions and the future studies. Suggest moving them to the main manuscript.

We have now quantified the proportion of cER decorating adhesive membranes and show these data in Fig 5C.

The proximity ligation experiments are not sufficient on their own and would need independent validation somehow to support the conclusion of the redistribution of IP3 receptors away from the plasma membrane, such as co-IP or immunogold localization.

To complement the PLA data, we have quantified the redistribution of IP₃ receptors by immunogold. Significantly fewer gold particles were detected in Stim1/2-deficient neutrophils, the total number of gold particles and those located <50 nm from the PM decreasing by 73% and 66 %, respectively (Fig. 6B). The mRNA levels of the three IP₃R isoforms were comparable in the two genotypes (Fig. S6E), suggesting that Stim1/2 deficiency post-translationally impacts IP₃R stability. These data are congruent with our PLA data showing a reduction of IP₃ receptors at ER-PM contact sites and indicate that Stim1/2 ablation impacts a core component of the Ca²⁺ signaling machinery.

Reviewer #2

This manuscript has several major flaws and the results do not warrant the conclusions summarized in the abstract.

We hope that our new experiments suitably address the flaws identified by the reviewer and that the additional evidence provided support our conclusions, which we have carefully edited to avoid unwarranted interpretation.

1. Figure 3. The role of Orai1 in generating calcium signals in neutrophils was not rigorously tested. The claim that calcium entry is not required for calcium activity of adherent neutrophils is based on the effect of only one Orai1 inhibitor.

We had shown lack of effect of two inhibitors, GSK7975A and Gd³⁺. In the revised version we show that another Orai1-specific inhibitor, CM4620, also fails to abrogate adhesion-associated Ca²⁺ transients in neutrophils (Fig. S3A). Together with the immediate and complete block observed with the PLC inhibitor U73122, these data provide strong evidence that the Ca²⁺ activity does not reflect Ca²⁺ entry, but Ca²⁺ released by IP₃ from intracellular stores.

2. Figure 5. The authors claimed that "ER-PM contact sites were dynamically excluded from actin-rich adhesive membranes in WT, but not Stim1/2-deficient neutrophils, linking the persistence of signaling-deficient contact sites to impaired actin remodeling". Nevertheless, the authors did not show or analyze "actin-rich adhesive membranes" in this manuscript. Also, the concept that ER-PM contact sites are excluded from actin-rich membrane is not new.

As suggested by reviewer 1 we have now documented extensively the impact of Stim1/2 ablation on the remodelling of the actin cytoskeleton, using light and electron microscopy. Stim1/2-deficient neutrophils had reduced phalloidin staining at adhesion sites, impaired SiR-actin dynamics during spreading, and a reduced actin-dense coverage of adhesive membranes on electron micrographs (Fig. 7). Furthermore, we show that increasing the external Ca²⁺ concentration from 2 to 10 mM restored basal Ca²⁺ levels, spontaneous Ca²⁺ activity, and spreading efficiency in Stim1/2-deficient neutrophils (Fig. 6). These data mechanistically link Stim1/2-dependent Ca²⁺ elevations to the formation of actin-rich adhesive membranes during neutrophil spreading. We agree that the concept that ER-PM contact sites are excluded from actin-rich membrane is not new, but a causal relationship between contact site formation and cell directionality during migration was only established recently (Gong et al., 2024). Our data provide molecular insight on this process by showing that Stim proteins can act as labile ER-PM tethers at adhesion sites during neutrophil spreading.

3. Figure 5. The authors emphasize that "unexpectedly (STIM1/2 ablation) did not impact the proportion, proximity, and length of ER-PM contact sites". However, many differences between WT and STIM1/2 DKO,

especially at the adhesive membrane where cell spreading occurs, were shown in Figure S5B and S5C. There is a bias toward the conclusion the authors would like to make.

We stressed this point because we expected Stim1/2-deficiency to have a major impact on contact site formation, as in cultured cell lines Stim1 ablation greatly reduces the proportion of cortical ER. Our electron micrographs are the first to document the ultrastructure of membrane contact sites in primary immune cells lacking Stim1/2. To avoid bias caused by cellular activation we compared neutrophils in suspension and were surprised by the conserved proportion and morphometric parameters of contact sites in Stim1/2-deficient cells. We believe that this unexpected result is important and deserves recognition as it shows that the lack of Stim proteins can be compensated by other tethers in neutrophils. Differences became apparent only when comparing the adhesive membranes of adherent cells, and we have rephrased this sentence to better reflect the ultrastructural differences between wild-type and Stim1/2-deficient neutrophils.

4. Figure 6A. It does not make sense to perform PLA experiments between S1-O1 or S1-IP3R in S1/S2 DKO cells. Also, more experiments are needed to rigorously test the hypothesis that "ER Ca²⁺ release channels are redistributed away from the PM" in STIM1/2 DKO cells in addition to counting dots per cell from IP3R-PMCA PLA experiments.

To complement the PLA data, we have quantified the redistribution of IP₃ receptors by immunogold. Significantly fewer gold particles were detected in Stim1/2-deficient neutrophils, the total number of gold particles and those located <50 nm from the PM decreasing by 73% and 66 %, respectively (Fig. 6B). The mRNA levels of the three IP₃R isoforms were comparable in the two genotypes (Fig. S6E), suggesting that Stim1/2 deficiency post-translationally impacts IP₃R stability. These data are congruent with our PLA data showing a reduction of IP₃ receptors at ER-PM contact sites and indicate that Stim1/2 ablation impacts a core component of the Ca²⁺ signaling machinery. We have moved to supplemental material the PLA experiments between S1-O1 or S1-IP₃R in S1/S2 DKO cells that were meant as controls for the specificity of the anti-STIM1 antibody.

5. Figure 6B. It has been shown previously that STIM2 deficiency results in reduced basal calcium (PMID: 18160041)

We now mention the study by Brandmann et al. reporting the impact of STIM2 deficiency on basal calcium levels. The lack of Stim2 might account for the reduced basal Ca²⁺ levels of Stim1/2-deficient neutrophils, and we are currently generating mice bearing a targeted disruption in the Stim2 gene to test this hypothesis. Our new data also show that increasing the external Ca²⁺ concentration to 10 mM rapidly restores the Ca²⁺ signaling and motility defects of Stim1/2-deficient neutrophils (Fig. 6C-D). This rescue goes hand in hand with a restoration of basal Ca²⁺ levels, hinting at a critical role for the basal cytosolic Ca²⁺ levels in the control of Ca²⁺-dependent actin-based motility.

Reviewer #3

Although the studies are generally sound and use state-of-the-art technological approaches, some of the experiments seem descriptive in nature.

We agree that our experiments are descriptive, but this is an intrinsic limitation of studying the cell biology of intact neutrophils, being from human or mouse. Neutrophils are terminally differentiated cells whose lifespan is only a couple of hours in vitro. To study neutrophil physiology involves the generation of genetically modified

mice and extraction of primary cells for short-term functional assays. By carefully reporting neutrophils signaling and spreading defects in mice expressing Salsa6f and lacking Stim1/2 we gained new insights on the role of Stim proteins that are relevant for cell biologists.

Furthermore, expansion of the sample size in some experiments and complementation with genotype reversal approaches (rescue experiments) would help solidify some of the concepts and interpretations.

We have expanded the sample size whenever possible and performed the suggested rescue experiments (Fig. 6C-D).

Major comments

1. myeloid-specific Stim1/2 ablation prevents adhesion-induced Ca²⁺ elevations and impairs the spreading of mouse neutrophils. at least one rescue experiment would help validate the role of STM1 or 2 in these processes.

We agree that rescue by protein re-expression will establish causality, but unfortunately primary neutrophils cannot be genetically manipulated. Instead, we attempted to rescue the Ca²⁺ signaling and spreading defects of Stim1/2-deficient cells by increasing the extracellular Ca²⁺ concentration to supraphysiological levels. Remarkably, increasing the external Ca²⁺ concentration to 10 mM restored within minutes the Ca²⁺ activity and spreading competence of Stim1/2-deficient neutrophils (Fig. 6C-D). These data mechanistically link the signaling and motility defects imparted by Stim1/2 ablation, providing a conceptual framework that explain the adhesion defect of these cells.

2. Using pharmacological inhibitors, the authors conclude that the inhibitory effect of Stim1/2 depletion on calcium/spreading is recapitulated by PLC inhibition, leading to the conclusion that "... Stim1/2 ablation impairs the PLC-dependent spreading of neutrophils". However, in Figure S4B the authors show that the PLC inhibitor U73122 inhibits spreading not only in WT but also in Stim1/2-KO cells suggesting that PLC regulates spreading by interfering with mechanisms independent of STIM1/2. This should be discussed.

We agree and have reworded this statement to "Stim1/2 ablation reduces the PLC-dependent spreading of neutrophils". Engagement of PLC-coupled adhesion and chemokine receptors generate multiple STIM-independent intracellular signals that can account for the effect of the PLC inhibitor in Stim1/2-deficient cells.

3. Proximity ligation assay is a powerful technique however, S1-O1 interaction is visualized in Stim1/2-KO, with many cells showing as many as 3 dots/cell. Does the author consider this to be the background level? For reference, a similar number of interactions are observed in WT cells for S1-IP3R. The authors should complement these studies with colocalization analyses of endogenous proteins in 3-D. Also, additional information should be added to the methods section, specifying that PLA was analyzed in 3-D as well as the step used in this analysis.

To complement the PLA data, we have quantified the redistribution of IP₃ receptors by immunogold. Significantly fewer gold particles were detected in Stim1/2-deficient neutrophils, the total number of gold particles and those located <50 nm from the PM decreasing by 73% and 66 %, respectively (Fig. 6B). The mRNA levels of the three IP₃R isoforms were comparable in the two genotypes (Fig. S6E), suggesting that Stim1/2 deficiency post-translationally impacts IP₃R stability. These data are congruent with our PLA data showing a reduction of IP₃ receptors at ER-PM contact sites and indicate that Stim1/2 ablation impacts a core component of the Ca²⁺ signaling machinery. As recommended by reviewer 2, we have moved to supplemental material the PLA experiments with S1-O1 or S1-IP₃R in Stim1/2-KO, meant as controls for the specificity of the anti-STIM1 antibody (and interpreted as background signals). We also provide additional information in the methods section, specifying that PLA was analyzed in 3-D and the steps used in this analysis.

4. Define the abbreviations in figure legends, for example, S1 in Figure 6. Done

5. In some experiments (for example Fig. 6 but also in many others), only "ten cells from 2 pairs are analyzed". The authors should expand the "n, (number of mice)" in key experiments and data presented as super-plots whenever possible.

We have expanded the sample size in key experiments, increasing the number of mice when allowed by the strong constraints imposed on animal experimentation in Switzerland. We present the data as super-plots whenever possible. Thank you for suggesting this transparent mode of presentation.

Minor comments:

1. The author should clearly indicate where they use post-hoc neutrophil identification or isolation of neutrophils by negative selection. Cells are indicated as "flushed" or "purified" in figure legends.

2. The authors wrote "Inspection of the electron micrographs indicated that WT neutrophils had a long adhesive membrane covered by a thick actin cytoskeleton that was largely devoid of contact sites (Fig. 5A)." This should be rephrased as the thick actin cytoskeleton is not visualized in these micrographs.

We provide additional electron micrographs illustrating the thickness of the actin cytoskeleton over the adhesive membranes (indicated in the images) and its quantification (Fig. 7).

3. The putative reduced immunoreactivity of IP₃R in S1/2Ko (Fig. S6B) is too mild and the use of n=1 mouse pair raises questions about whether this difference is real. Expanding the n and using super-plots would help establish whether there are biological differences in IP₃R levels.

We now show data obtained by immunogold in another mouse pair that confirm a decrease in IP₃R immunoreactivity in Stim1/2-deficient cells (Fig. 6B). Our new PCR data, however, indicate preserved expression levels of the three IP₃R isoforms (Fig. S6D). This suggests that Stim1/2 ablation might decrease the expression of IP₃R post-translationally, possibly by preventing the stabilization of licensed receptors at ER-PM contact sites.

References cited:

Gong, B., J.D. Johnston, A. Thiemicke, A. de Marco, and T. Meyer. 2024. Endoplasmic reticulum-plasma membrane contact gradients direct cell migration. *Nature*. 631:415-423.

Jaconi, M.E., J.M. Theler, W. Schlegel, R.D. Appel, S.D. Wright, and P.D. Lew. 1991. Multiple elevations of cytosolic-free Ca²⁺ in human neutrophils: initiation by adherence receptors of the integrin family. *J Cell Biol*. 112:1249-1257.

January 14, 2025

RE: JCB Manuscript #202406053R

Nicolas Demaurex
University of Geneva

Dear Prof. Demaurex:

Thank you for submitting your revised manuscript entitled "STIM1/2 maintain signaling competence at ER-PM contact sites during neutrophil spreading," and for your patience during the long duration of the re-review process. We would be happy to publish your paper in JCB pending the resolution of remaining reviewer concerns, and final revisions necessary to meet our formatting guidelines (see details below).

As you will see, reviewers all appreciated the significant effort made to improve this manuscript which strengthen the observations that relate calcium signaling through STIM1/2 to IP3R and neutrophil spreading. However, reviewers remarked that immunogold measurements of IP3R are somewhat in conflict with the model set forth, and reviewers pointed to specific claims made in the text that must be adjusted to better align with the new data. In this regard, Reviewer 1 proposed the possibility that IP3R protein levels are repressed globally in STIM1/2 KO neutrophils and requested western blotting to measure this. We agree this work must include these data, which will be valuable in interpreting the effects of STIM1/2 loss. In addition, Reviewer 3 made important suggestions regarding how immunogold data are presented, and we strongly encourage you to increase n and/or discuss the limitations of this method in the text. Finally reviewers noted the surprising results from intravital imaging, which we also found intriguing although not necessarily in conflict with deficient cell spreading and adhesion seen in culture. We will evaluate a revised manuscript incorporating the above changes without further reviewer input.

A. MANUSCRIPT ORGANIZATION AND FORMATTING:

Full guidelines are available on our Instructions for Authors page, <http://jcb.rupress.org/submission-guidelines#revised>. Submission of a paper that does not conform to JCB guidelines will delay the acceptance of your manuscript.

- 1) Text limits: Character count for Articles is < 40,000, not including spaces. Count includes abstract, introduction, results, discussion, and acknowledgments. Count does not include title page, figure legends, materials and methods, references, tables, or supplemental legends.
- 2) Figures limits: Articles may have up to 10 main figures and 5 supplemental figures/tables.
- 3) Figure formatting: Scale bars must be present on all microscopy images, including inset magnifications. Molecular weight or nucleic acid size markers must be included on all gel electrophoresis. Please avoid pairing red and green for images and graphs to ensure legibility for color-blind readers. If red and green are paired for images, please ensure that the particular red and green hues used in micrographs are distinctive with any of the colorblind types. If not, please modify colors accordingly or provide separate images of the individual channels.
- 4) Statistical analysis: Error bars on graphic representations of numerical data must be clearly described in the figure legend. The number of independent data points (n) represented in a graph must be indicated in the legend. Statistical methods should be explained in full in the materials and methods. For figures presenting pooled data the statistical measure should be defined in the figure legends. Please also be sure to indicate the statistical tests used in each of your experiments (either in the figure legend itself or in a separate methods section) as well as the parameters of the test (for example, if you ran a t-test, please indicate if it was one- or two-sided, etc.). Also, if you used parametric tests, please indicate if the data distribution was tested for normality (and if so, how). If not, you must state something to the effect that "Data distribution was assumed to be normal but this was not formally tested."
- 5) Abstract and title: The abstract should be no longer than 160 words and should communicate the significance of the paper for a general audience. The title should be less than 100 characters including spaces. Make the title concise but accessible to a general readership.
- 6) Materials and methods: Should be comprehensive and not simply reference a previous publication for details on how an experiment was performed. Please provide full descriptions in the text for readers who may not have access to referenced manuscripts. We also provide a report from SciScore and an associate score, which we encourage you to use as a means of evaluating and improving the methods section.

7) Please be sure to provide the sequences for all of your primers/oligos, plasmids, and RNAi constructs in the materials and methods. You must also indicate in the methods the source, species, and catalog numbers (where appropriate) for all of your antibodies. Please also indicate the acquisition and quantification methods for immunoblotting/western blots.

8) Microscope image acquisition: The following information must be provided about the acquisition and processing of images:

- a. Make and model of microscope
- b. Type, magnification, and numerical aperture of the objective lenses
- c. Temperature
- d. Imaging medium
- e. Fluorochromes
- f. Camera make and model
- g. Acquisition software
- h. Any software used for image processing subsequent to data acquisition. Please include details and types of operations involved (e.g., type of deconvolution, 3D reconstitutions, surface or volume rendering, gamma adjustments, etc.).

10) Supplemental materials: There are strict limits on the allowable amount of supplemental data. Articles may have up to 5 supplemental figures. Please also note that tables, like figures, should be provided as individual, editable files. A summary of all supplemental material should appear at the end of the Materials and methods section.

13) ORCID IDs: ORCID IDs are unique identifiers allowing researchers to create a record of their various scholarly contributions in a single place. At resubmission of your final files, please provide an ORCID ID for all authors.

15) A data availability statement is required for all research article submissions. The statement should address all data underlying the research presented in the manuscript. Please visit the JCB instructions for authors for guidelines and examples of statements at (<https://rupress.org/jcb/pages/editorial-policies#data-availability-statement>).

Please note that JCB requires authors to submit Source Data used to generate figures containing gels and Western blots with all revised manuscripts. This Source Data consists of fully uncropped and unprocessed images for each gel/blot displayed in the main and supplemental figures. Since your paper includes cropped gel and/or blot images, please be sure to provide one Source Data file for each figure that contains gels and/or blots along with your revised manuscript files. File names for Source Data figures should be alphanumeric without any spaces or special characters (i.e., SourceDataF#, where F# refers to the associated main figure number or SourceDataFS# for those associated with Supplementary figures). The lanes of the gels/blots should be labeled as they are in the associated figure, the place where cropping was applied should be marked (with a box), and molecular weight/size standards should be labeled wherever possible. Source Data files will be directly linked to specific figures in the published article.

Journal of Cell Biology now requires a data availability statement for all research article submissions. These statements will be published in the article directly above the Acknowledgments. The statement should address all data underlying the research presented in the manuscript. Please visit the JCB instructions for authors for guidelines and examples of statements at (<https://rupress.org/jcb/pages/editorial-policies#data-availability-statement>).

B. FINAL FILES:

Please upload the following materials to our online submission system. These items are required prior to acceptance. If you

have any questions, contact JCB's Managing Editor, Lindsey Hollander (lhollander@rockefeller.edu).

Thank you for your attention to these final processing requirements. Please revise and format the manuscript and upload materials within 7 days. If you need an extension for whatever reason, please let us know and we can work with you to determine a suitable revision period.

Thank you for this interesting contribution, we look forward to publishing your paper in Journal of Cell Biology.

Sincerely,

Tamas Balla
Monitoring Editor
Journal of Cell Biology

Tim Fessenden
Scientific Editor
Journal of Cell Biology

Reviewer #1 (Comments to the Authors (Required)):

I thank the authors for their responses and the additional experiments performed. The new data are quite interesting and provide additional insights, especially the rescues observed by simply placing the cells in high Ca²⁺. Furthermore, it is important to point out the technical difficulties involved in working with neutrophils as pointed out by the authors. However, the new results raise important issues in terms of the significance of the elegant cell biological findings physiologically. In addition, the conclusions regarding the role IP3Rs are not fully justified as discussed below.

1. The in vivo phenotype of increased migration of neutrophils to the site of infection is opposite to what would have been expected from the in vitro phenotype of altered basal Ca²⁺, cER, actin dynamics, junctions, spreading and migration. This argues that observed phenotypes in vitro are not primary drivers of neutrophil behavior in vivo bringing into question their physiological relevance.

2. The conclusions in the revised manuscript regarding the role of IP3Rs are not justified by the presented data. The statements: "Our findings indicate that Stim1/2 stabilize IP3Rs at ER-PM nanojunctions" (abstract); "controlling... the dynamic delivery and retrieval of IP3Rs-containing cER sheets at sites of receptor engagement" (conclusion) seem at odd with the data presented in Fig. 6B where there is less IP3Rs detected by immunogold staining globally in the cell, arguing for decreased IP3R protein levels and not their stability at the junctions or delivery/retrieval. This would explain the decreased PMCA interaction in the PLA

assay and cER presence. So mechanistically loss of Stim1/2 in neutrophils may be regulating IP3R protein levels and not their distribution. Did the authors perform a WB or immunofluorescence to check IP3R protein levels? This is an important point to clarify as it will drastically affect the conclusions reached.

3. Similar comment for Junctate and eSyt2 levels using RT-PCR, can those be confirmed at the protein level?

Reviewer #2 (Comments to the Authors (Required)):

The revised manuscript provides new and interesting results. Fig. 6B showed that there is an overall decrease in IP3R proteins in Stim1/2-deficient neutrophils, not specifically at ER-PM contact sites as claimed in the original manuscript. The authors should change their claims about IP3R accordingly.

Specifically, the statement about IP3Rs in the Abstract "Stim1/2 stabilize IP3Rs at ER-PM contacts" should be changed to something like "Stim1/2 maintain IP3Rs in neutrophils".

Also, the statement about IP3Rs in the Summary Statement "Stim1/2 deficiency depletes IP3Rs from the plasma membrane without reducing membrane contact sites" should be changed since the results showed that Stim1/2 deficiency depletes IP3Rs everywhere in neutrophils, not just "from the plasma membrane".

Moreover, the extensive discussion on the redistribution of IP3Rs away from the PM in the Discussion session should be removed or modified.

Lastly, new results shown on Fig. 8 (in vivo studies) are interesting but puzzling. It would be helpful to include discussions on the increased recruitment of Stim1/2-deficient neutrophils to sites of inflammation based on findings in Figures 1 to 7 and in previous literature.

Reviewer #3 (Comments to the Authors (Required)):

The manuscript is greatly improved, and the authors have made considerable efforts to answer this reviewer's and other reviewers' concerns.

Improvements include:

1) A rescue analysis using supraphysiologic calcium was included, showing PLC dependence of this effect using U73122. (By the way, mouse neutrophils can be transfected.)

2) Quantification of IP3R immunoreactivity in proximity to the PM in Stim1/2-deficient neutrophils by immunogold. (but see comment below)

3) The inclusion of super-plots increases transparency.

However, there is still room for improvement:

1) Immunogold quantification is performed on 1 wt and 1 kO mice. The representative image is only a WT cell, but KO cells are not shown. In the image, while the cell indicated with arrows clearly shows 3 gold particles near the PM, other cells in the same field do not. In this type of assay, only an 80-nm thick layer of a cell is visualized, stressing the need to increase the n, showing more cells and including images of the KO cells.

2) The EM shown in Fig. 7A makes the assumption that all the area lacking organelles/nuclei in adherent cells is occupied by actin. Although differences between WT and KO in this "empty space" is clear, unless cells are labeled for actin, the word "actin" should be excluded from the micrographs and the text should be reworded accordingly. This is attenuated by the analysis of sir-Act. Still, one cannot be sure that all the space in EM images is occupied by actin.

Geneva, January 31, 2025

Dear Editors,

We were pleased to read that our manuscript could be published in JCB pending the resolution of remaining reviewer concerns. As suggested, we have performed western blots and analyzed additional immunogold micrographs to assess IP₃R levels. Interestingly, the amounts of the two main isoforms, IP₃R1 and IP₃R3, were similar in WT and KO cells on Western blots while IP₃R1 immunoreactivity was decreased on confocal and immunogold micrographs. This suggests that Stim1/2 deficiency does not reduce IP₃R1 levels but modifies the conformation of the receptor to a less immunoreactive form in cells. We have added these new data (new Fig 6B, 6C, and S5B) and amended our manuscript to incorporate these findings and the reviewers' suggestions.

The changes are detailed in the point-by-point responses to reviewers' comments (in *italics*) below.

Reviewer #1 (Comments to the Authors (Required)):

1. Physiological relevance of the in vitro phenotypes in view of the increased migration in vivo.

We were also expecting a reduced migration in vivo but disagree that the phenotypes that we report in vitro are not relevant. In vivo recordings reflect the net result of the multi-step migration process occurring under flow conditions, while in vitro recordings explore only the initial adhesion steps in static conditions. Physiological insight is gained by combining both approaches. In our case, the reduced spreading of Stim1/2-deficient neutrophils in vitro might translate into a less adhesive phenotype in vivo. This could enhance the intraluminal crawling of KO neutrophils and facilitate their transmigration by allowing cells to find optimal emigration sites more efficiently. Moreover, our findings that Stim1/2-dependent defects are alleviated at high Ca²⁺ concentrations might explain contradicting reports on the impact of Stim1/2 ablation in vivo. Normal neutrophil recruitment was observed in models of established peritoneal and subcutaneous inflammation (Clemens et al., 2017; Zhang et al., 2014) and reduced recruitment in imiquimod-induced psoriasis (Steinckwich et al., 2015). In most tissues neutrophils are exposed to 1-2 mM but the Ca²⁺ concentration decreases to 0.2-0.4 mM in the mammalian epidermis (Elias et al., 2002). These low external Ca²⁺ concentrations might exacerbate Stim1/2-dependent defects, accounting for the reduced recruitment of Stim1/2-deficient neutrophils in the inflamed skin. These issues are now discussed in the revised MS.

2. Stim1/2 in neutrophils may be regulating IP3R protein levels and not their distribution.

We agree with this comment and have assessed IP₃R protein levels by WB and immunofluorescence. Interestingly, the amounts of the two main isoforms, IP₃R1 and IP₃R3, were similar in WT and Stim1/2-deficient neutrophils on Western blots while IP₃R1 immunoreactivity was decreased on histochemical and immunogold micrographs. We have added these new data (new Fig 6B, C and S5B) and amended our manuscript to incorporate these findings and the reviewers' suggestions.

3. Are junctate and eSyt2 also decreased at the protein level? We did not perform these validations that would have delayed the publication of our provisionally accepted manuscript-

Reviewer #2 (Comments to the Authors (Required)):

As discussed above, the levels of the two main IP₃R1 and IP₃R3 isoforms measured by Western blot were similar but IP₃R1 immunoreactivity was decreased on confocal and immuno-EM micrographs of

Stim1/2-deficient neutrophils. This suggests that Stim1/2 deficiency does not reduce IP₃R1 levels but modifies the conformation of the receptor to a less immunoreactive form in cells. We have added these new data (new Fig 6B, 6C, and S5B) and amended our text to acknowledge this unexpected finding and to discuss the increased recruitment of Stim1/2-deficient neutrophils to sites of inflammation in view of earlier findings.

Reviewer #3 (Comments to the Authors (Required)):

1) Increase the n, showing more cells and including images of the KO cells in immunogold data.

We have analyzed additional immunogold micrographs from different grids, increasing n to 2x40 cells in two biological replicates. The new Fig. 6C shows images of both WT and KO cells with insets to illustrate the global and cortical immunoreactivity.

2) Avoid labeling the empty space "actin" in the EM pictures of Fig 7A as one cannot be sure that all the space in EM images is occupied by actin.

We have changed the labels on the micrographs and reworded the text accordingly.

Hoping that our study is now suitable for publication in JCB

Sincerely

Nic Demaurex
